

# Poro-perm relations of Mesozoic carbonates and fault breccia, Araxos Promontory, NW Greece

Sergio C. Vinciguerra[1], Federico Vagnon[2], Irene Bottero[1], Jerome Fortin[3], Angela Vita Petrullo[4],

Dimitrios Spanos[5], Aristotelis Pagoulatos[5], Fabrizio Agosta[4,6]

[1]Department of Earth Sciences, University of Turin, Turin, 10125, Italy

[2]Department of Environment, Land and Infrastructure Engineering, Politecnico di Torino, Turin, 10129, Italy

[3]Laboratoire de Géologie - Ecole Normale Supérieure/ CNRS UMR8538, PSL Research University, Paris, 75005, France

[4]GeoSMART Italia s.r.l.s, Potenza, 85100, Italy

[5] Hellenic Petroleum, Athens, 15125, Greece

[6] Department of Science, University of Basilicata, Potenza, 85100, Italy

*Correspondence to*: Sergio C. Vinciguerra (sergiocarmelo.vinciguerra@unito.it)

**Abstract.** Aiming at assessing the porosity and permeability properties, we present the results of microstructural and laboratory measurements, including density, porosity, $V_P$, $V_S$, and electrical resistivity. These measurements were performed in dry and in saturated conditions on 54 blocks of Mesozoic carbonate host rocks and fault breccias collected in Greece. The host rocks consist of carbonate mudstones, wackestones, packstones, and sedimentary breccias from the Senonian and Vigla formations. These rocks exhibits average density values, low porosity values, and medium-to-high P- and S-wave velocities. Fault breccias originate from high-angle extensional and strike-slip fault zones, displaying a wider range of density, porosity values up to 5-10 times higher than host rock, along with ultrasonic velocities. Regardless of lithology, the carbonate host rocks might include vugs due to selective dissolution. Conversely, the fault breccia samples feature microfractures. Slight textural anisotropy is documented in the carbonate host rocks, while a higher degree of anisotropy characterizes the fault breccias. Selected samples were also tested in pressure vessels with confining pressure up to 80 MPa, revealing that transport properties along microcracks in fault breccias can significantly increase with increasing depth. To assess rock permeability and porosity-permeability relations, three different protocols were employed. Two of them were based on the Effective Medium Theory, where permeability was computed by inverting ultrasonic measurements, assuming an array of penny-shaped cracks embedded in an impermeable host matrix. The aspect ratio and crack width were obtained by the seismic measurements, modeling either by assuming all cracks as isolated or unconnected or all cracks connected into the network. The application of these two protocols showed a systematic variation of permeability with porosity. In contrast, the results of the third protocol, based on the digital image analysis outcomes only, did not exhibit systematic variation. This behavior was interpreted as a result of the not-selective dissolution of the outcropping carbonates causing a wide range of measured fracture aperture values. This study found that carbonate host rocks lacked a clear poro-perm trend due to the presence of stiff, sub-rounded pores and small vugs. On the contrary, fault breccia exhibited a linear increase in permeability with porosity due to a connected pore network including microfractures.



## 1 Introduction

Geophysical exploration methods commonly employed for the analysis of potential geofluids reservoirs include seismic survey, gamma ray logging, stress field detection survey, controlled source electromagnetic data surveys and many others (Unz, 1959; Wang et al., 2021). Commonly, the petrophysical analyses are performed on well logs or on undisturbed core samples directly retrieved from wells. However, prior to subsurface investigations, laboratory and on-field measurements are performed on outcropping analogs of deep reservoirs to identify fractures, faults and to assess their role on transport properties (Simmons and Cooper, 1978; Walsh, 1965). It is understood that isolated fractures such as cracks (referred to as joints for geologists) make rocks more compliant, while connected cracks make rocks more permeable and anisotropic when aligned to each other (Nelson, 2001). The superposition of multiple deformation mechanisms over extended periods of time may give rise to a complex rock fabric, thereby making the interpretation of field-scale seismic surveys highly challenging unless a priori knowledge of the physical state of the rock is established and understood. Moreover, the rock fabric is also affected by the diagenetic processes (Laubach et al., 2010; Bailly et al. 2019a,b, 2022), particularly in carbonates, where physical/chemical compaction and pervasive cementation occur (Ferraro et al. 2019).

At a larger scale, fault zones in cemented carbonates may exert significant control on both storage and transport fluid properties (Ferraro et al., 2020; Chicco et al. 2023). Fault zones often include fault cores (FCs) consisting of fault rocks and main slip surfaces, as well as fault damage zones (DZs) consisting of fractured carbonates crosscut by small, subsidiary faults (Agosta and Aydin, 2006; Giuffrida et al., 2019). These zones form combined barrier-conduit permeability structures (Caine et al., 1996), where low permeability FCs are flanked by fault zones (FZs), enhancing fault-parallel fluid flow (Rawling et al., 2001; Ferraro et al., 2018; Volatili et al., 2022). In the case of un-cohesive and permeable fault rocks, FCs also conduce fluids allowing entire fault zones to function as distributed fluid conduits (Caine et al., 1996; Agosta et al., 2021).

However, estimating the permeability of both FCs and DZs is a complex task due to the heterogeneity and anisotropy of rock masses (Piscopo et al., 2018). Consequently, a large number of theoretical models has been proposed to predict changes in one property of cracked rocks based on the measurement of another property (Benson et al., 2006). Nonetheless, to apply any of these models, it is necessary to verify them by performing systematic control experiments in which all the relevant properties are measured on the same sample under the same experimental conditions. This approach eliminates possible errors arising from depositional and structural heterogeneities that form prominent anisotropies within carbonate rocks. To date, such systematic studies have been infrequent.

In this work, we focus on the petrophysical characterization of rock samples collected from the Araxos Promontory, in western Greece. This region, characterized by the presence of major petroleum systems extending into the Ionian Sea (Karakitsios, 2013), has been the subject of recent studies documenting the sedimentary infill of the Ionian Basin (Bourli et al., 2019a, b), and its tectonic evolution (Tavani et al., 2019, Smeraglia et al., 2023). Aiming to define the poro-perm relations for both host and fault rocks, we investigate a variety of carbonate lithofacies by integrating the results of 2D image and petrophysical analyses to compute permeability values and compare them with experimental measurements. Results are discussed in light of the direct observation of connected pore space, presenting a protocol for petrophysical carbonate analysis. The findings have potential applications in many geo-engineering fields, including $CO_2$ storage in fractured carbonate reservoirs, geothermal energy production from low-middle enthalpy sites, and the preservation of deep aquifers during fossil energies production.



## 2    Geological setting

At a larger scale, the study area of the Araxos Promontory is part of the Dinarides-Albanides-Hellenides orogenic belt, which formed as a consequence of the Cenozoic collision between the Adria-Africa and Eurasian continental plates (Papanikolaou, 2021; Robertson and Shallo, 2000; Roure François et al., 2004; Underhill, 1989). The Hellenides Fault ant Thrust Belt (FTB) is made up of four main structural domains, which respectively consist of from west to east (Figure 1a): (i) Apulian carbonate platform, (ii) Ionian pelagic basin, (iii) Gavrovo carbonate platform, and (iv) Pindos oceanic basin (Karakitsios, 2013b; Underhill, 1989). These structural domains were tectonically juxtaposed during late Eocene-Miocene piggy-back thrusting tectonics (Robertson and Dixon, 1984) with occasional out-of-sequence thrusting (Sotiropoulos et al., 2003). Currently, the outer domain of the Hellenides FTB is subjected to an active compression (Kiratzi and Louvari, 2003; Jolivet and Brun, 2010 ).

The Araxos Promontory pertains to the Internal Ionian Zone, east to the Gavrovo thurst (Figure 1a). It is bounded northward by the roughly WNW striking and seismically active normal faults of the Gulf of Patras and Corinth (Jolivet et al., 2015), and flanked by Holocene continental deposits both eastward and southward. The studied area is made up of an E-SE dipping carbonate monocline including Lower Cretaceous, chert-bearing, deep-water limestones (Danelian et al., 2018; Skourtsis-Coroneou et al., 1995), and Upper Cretaceous-Eocene pelagic and hemipelagic limestones with frequent calciturbidites and sedimentary breccias interlayers (Bourli et al., 2019a; Karakitsios and Rigakis, 2007). The latter breccias and calciturbidites are made up of Vigla-derived clasts, and with minor intercalations of thinly bedded pelagic limestones. A detailed lithostratigraphic representation of the Ionian carbonates is reported in Figure 1b (Karakitsios, 2013). Stampfli (2005) assessed that the Triassic-early Jurassic succession is also part of the syn-rift sequence. Altogether, the Triassic-early Jurassic pre-rift, the early Jurassic-Late Jurassic syn-, and the younger post-rift sequences were later deformed during thrusting tectonics by mean of NNW-SSE striking thrust faults and folds, which solved a WSW-ENE shortening direction (Tavani et al., 2019; Underhill, 1989). Smaller scale folds associated to N-S shortening were also documented east of the Lefkada Island, throughout the Hellenides, as well as in the Araxos Promontory (Smeraglia et al., 2023).



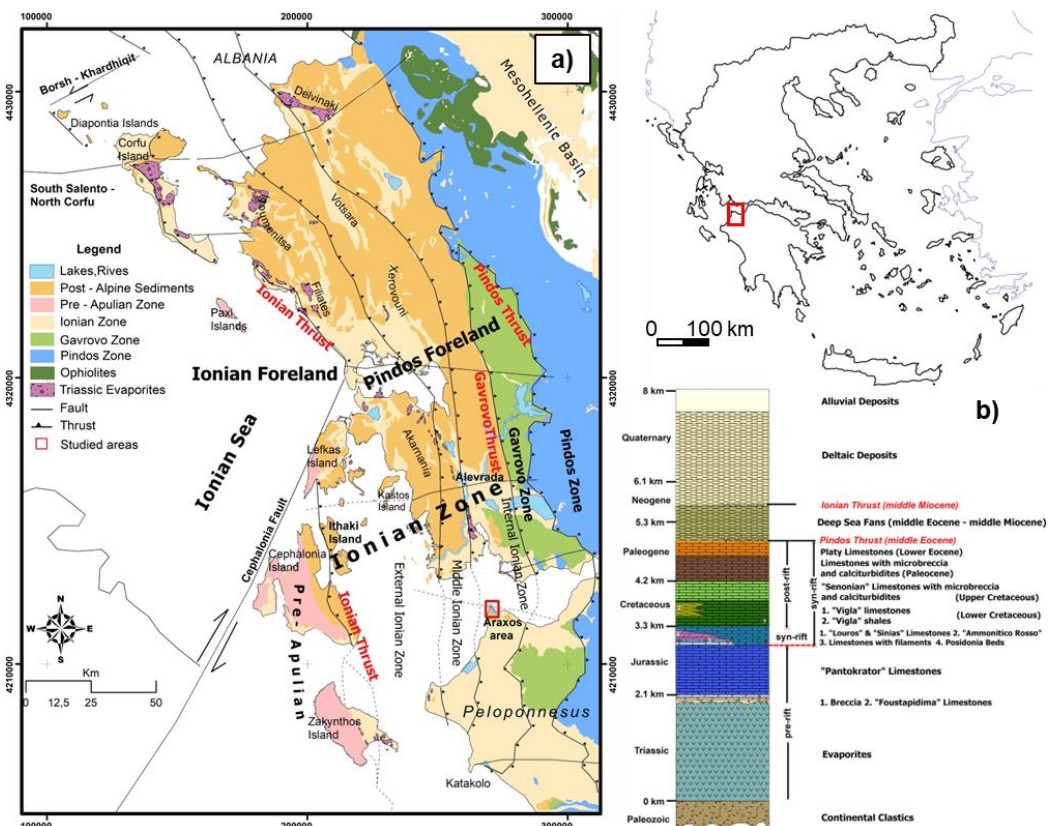

Figure 1: a) Geological sketch of the Western Greece and b) lithostratigraphic column of the Ionian zone (modified from (Bourli et al., 2019).

## 3 Methods

### 3.1 Sampling strategy and classification

Six to ten cm-sided cubes and parallelepipeds, hereafter named as rock blocks, were saw-cut from the oriented hand specimens collected in the field. The rock blocks were used to perform experimental analyses aiming at measuring the values of porosity, density, ultrasonic velocities and electrical resistivity at room pressure and temperature.

The samples were collected in the same study area along different fault zones, characterized by different kinematics (normal vs strike-slip), ages, and dimensions. The chosen hand specimens derive form the following structural domains (Woodcock and Mort, 2008) :

- 17 Host Rocks exposed Away from major Fault Zones (HR-AFZ);
- 7 Host Rocks in Proximity to Fault Zones (HR-PFZ);
- 5 Fragmented Host Rocks from Fault Zones (FHR-FZ);
- 10 Crush Fault Breccia from Fault Zones (CFB-FZ);
- 15 Fine Crush Fault Breccia and fault microbreccia from Fault Zones (FCFB-FZ).

The collected blocks were oriented with reference to the north and perpendicular to the stratification or fault direction, depending on whether they belonged to the host rocks or fault breccias.



### 3.2 Microstructural analyses

For rock texture digital analysis, 48 rock slabs obtained from the same hand specimens employed for petrophysical analyses were selected for microstructural analysis. In details:

- 6 host rock samples collected either away (HR-AFZ) from or in proximity to fault zones (HR-PFZ);
- 5 fragmented host rock samples (FHR-FZ) collected from fault zones;
- 19 crush fault breccia samples (CFB-FZ) from fault zones;
- 18 fine crush fault breccia and fault microbreccia samples (FCFB-FZ) collected from fault zones.

The rock slabs were saw cut either parallel or perpendicular to bedding/slip surfaces. Each rock slab was manually polished in order to later acquire good digital images, and then scanned by using high-resolution, 2400 dpi scanner. As a result, a detailed textural analysis was conducted aiming at assessing the main rock textures, as well as the shape, size, sphericity, and roundness of both clasts (host rock samples) and survivor grains (fault-related samples). Afterward, on the basis of a preliminary qualitative analysis, a discrete 1 $cm^2$-wide area was selected from the individual textures documented in single rock slabs to perform quantitative image analysis. This means that rock slabs with homogeneous texture were analyzed by selecting one representative area; those with multiple textures were analyzed by selecting two or more representative areas. In all cases, cm-sized clasts/grains were avoided. The digital images were processed by using both Gimp and ImageJ software applications, which allow to manually draw the individual clasts/survivor grains in order to carefully trace their individual shapes. As a result, a bitmap image was obtained for each 1$cm^2$ area; there, clast/survivor grains were shown in black, while both carbonate matrix and cement were shown in white (Figure 2a). Pores were not considered at this stage of the work. Afterwards, using the command "Analyze Particles" of the ImageJ software, all clasts/survivor grains were automatically processed to measure the following parameters: (i) particle count, (ii) total particle area, (iii) average particle size, (iv) area fraction representative of the percentage of clasts/survivor grains (pixels highlighted in black using the Threshold command), (v) percentage of matrix, and (vi) perimeter of the clast/survivor grains. Here we focused on the computation of the box-counting Dimension $D_{0(grains)}$ (Figure 2b). The resulting plots of box size vs. number of counts of the boxes with, at least, one portion of a clast/survivor grain. Data points were fitted by a power-law function (Mandelbrot, 1985; Falconer, 2003), and the angular coefficient of the best fit line represented the $D_0$ value. Specifically, $D_0$ is equal to the ratio between log N (N: number of boxes including the given object) and log r (r: inverse of the box size). We note that this digital image analysis is quite common in the case of cohesive rocks (Ferraro et al., 2018), when the traditional sieving technique cannot be performed.



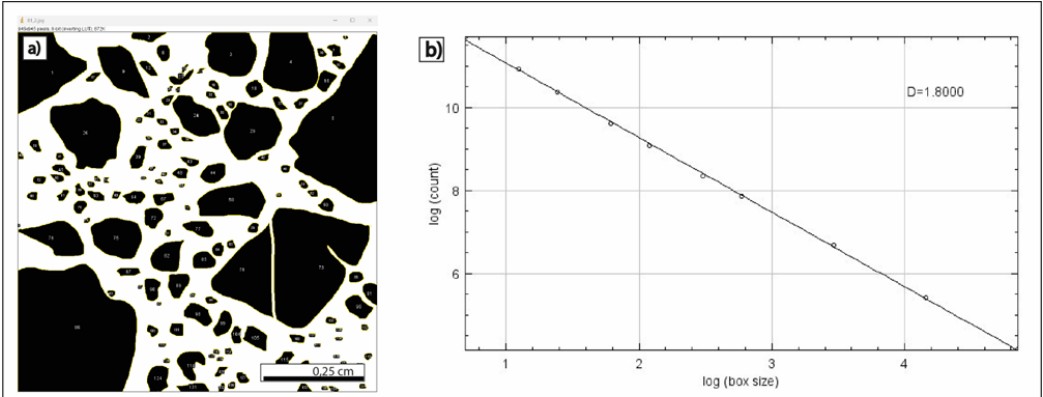

**Figure 2: a) Example of scanning of a polished plate, in which a representative, homogeneous area of 1 cm² is highlighted in the enlargement. In the corresponding bitmap image of the enlargement, the surviving grains or clasts are black, while the carbonate matrix, cements and pores are white. b) Example of a box size vs. number of counts plot, in a log-log space, which allows computation of the box-counting fractal dimension, $D_0$, which represents the angular coefficient of the best fit line. $D_0$ is therefore equal to the ratio between log N (number of boxes including the given object) and log r (inverse of the box size). We note that the box-counting fractal dimension, $D_0$, is automatically reported as D by the ImageJ software.**

Aiming at assessing the 2D pore properties, a 4 x 6.5 cm-wide thin section impregnated with a blue-dyed epoxy resin was then obtained from the single 48 rock slabs. Images of 6.17 x 4.55 mm-wide areas selected from individual thin sections were taken by means of a Nikon Eclipse E600 optical microscope equipped with a Nikon E4500 camera. Areas were chosen to fully characterize the textural heterogeneities observed in the thin sections. Areas containing macropores were excluded to prevent any potential bias in the calculated values. Afterward, applying the "select color" command of the Gimp software, the blue portions filled with the epoxy resin were selected. The obtained bitmap images highlighted the pores, shown with a black color, with respect to the clasts/survivor grains/matrix and cement shown with a white color. Then, by using the command "Analyze Particles" of ImageJ software (cf. Figure 3), all black objects (pores) included in the 6.17 x 4.55 mm-wide binary images were automatically counted to compute: (i) 2D connected porosity (percentage of pixels highlighted in black by using Threshold), (ii) circularity (shape factor, with 1.0 indicating a perfect circle and values close to zero indicating a very elongated shapes), and (iii) aspect ratio (ratio between longest and shortest axes of ellipses inscribed within each pore). The box-counting dimension, $D_{0(pores)}$ was then computed. This dimension had therefore a physical meaning similar to that of the fractal dimension related to pore throat distributions (Thompson, 1991; Ferraro et al., 2020).



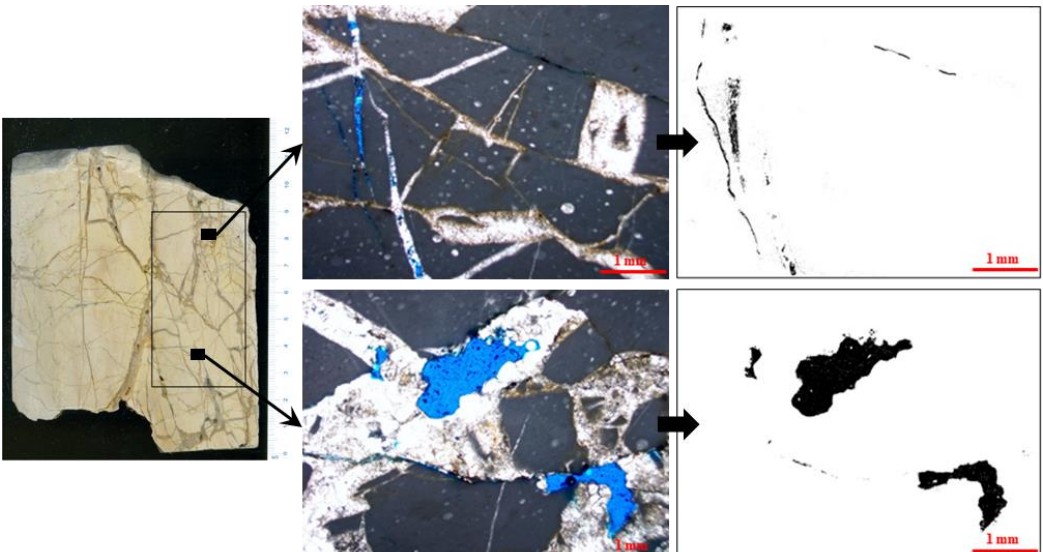

**Figure 3: Example of thin sections, in which two representative homogeneous areas of 1 cm² are highlighted in the**
**enlargements. In the corresponding bitmap image of the enlargement, pores are black, while clast, survivor grains, rock**
**fragments, matrix and cements are white.**

### 3.3 Petrophysical characterization at room pressure and temperature

#### 3.3.1 Dimensions, weight and volume

The measurement of physical properties such as density, ρ, and porosity, n, provides information on the mechanical
characteristics of rocks. The presence of pores in the rock fabric decreases its strength, and increases its deformability.
Carbonates usually exhibit a wide range of porosities, and hence various mechanical properties. Since the tested rock
blocks were considered with a regular geometry, and consisted of non-friable and coherent rocks (not appreciably swelling
or disintegrating when immersed in water), density and porosity were determined following the "Suggested methods for
determining water content, porosity, density, absorption and related properties" of (ISRM, 1979). With this aim, the bulk
volume, V, of each specimen was calculated by using the water displacement method with a precision of $10^{-4}$ m³. The
specimens were saturated by water immersion and repeated shaking (to removing trapped air) for one week. Tap water
with an electrical conductivity of 300 µS/cm was used as the saturating fluid. This choice was preferred over distilled
water, as the latter typically exhibits very low electrical conductivity values, which could influence the electrical
resistivity measurements performed on the same rock blocks (see section 3.3.3). The aforementioned methodology was
previously employed by the authors (Vagnon et al. 2019), yielding satisfactory results comparable with other independent
rock porosity measurements, even in rock with low porosity values. The saturated-surface-dry mass, $M_{sat}$, was then
determined by drying the surface with a moistened cloth, taking care to remove only the surface water, and weighting the
samples. The grain mass, $M_s$, was evaluated in natural condition without drying process in oven in order to avoid any
thermal alterations. Porosity and density (under dry, $\rho_{dry}$, and saturated, $\rho_{wet}$, sample conditions) were then obtained by
employing the following equations (1) to (4):

$$\rho_{dry} = \frac{M_s}{V} \qquad (1)$$



$$\rho_{wet} = \frac{M_{sat}}{V} \qquad (2)$$


$$n(\%) = \frac{100 \cdot V_v}{V} \qquad (3)$$

$$V_v = \frac{M_{sat} - M_s}{V} \qquad (4)$$

### 3.3.2 Ultrasonic Pulse Velocity (UPV) measurements

UPV measurements were performed using an ultrasonic pulse generation and acquisition system (Pundit Lab, Proceq). Two Cylindrical 250-kHz tx-rx probes transmitter-receiver (tx-rx) transducers were used for P-wave ($V_P$) and S-wave ($V_S$) measurements, along the three orthogonal directions of each oriented block. Measurements were conducted following (ASTM D2845-08, 2008) standard requirements (2008). For each sample, 11 ultrasonic traces were recorded, using a sampling frequency of 2 MHz. Manual picking of the first arrival times of the P-waves and S-waves was

performed. The determination of the P- and S- wave ultrasonic velocity was then straightforward as the dimensions of each block were previously measured. The representative velocity of each sample was chosen as the average of the measurements along the three directions. By employing the following equations (5 to 7), the combined measurement of both P- and S-wave velocities allowed us to retrieve the low-deformation (initial deformation phase of strain-stress curve) dynamic elastic parameters (Young modulus, E, shear modulus, G, and Poisson's ratio, ν) from indirect and non-

destructive tests:

$$G = \rho \cdot V_S^2 \qquad (5)$$

$$\nu = \frac{V_P^2 - 2V_S^2}{2(V_P^2 - V_S^2)} \qquad (6)$$


$$E = 2G(1 + \nu) \qquad (7)$$

### 3.3.3 Electrical Resistivity (ER) measurements

ER measurements were carried out with a measuring quadrupole connected to a Syscal-Pro (@Iris instruments) acquisition system. The instrumentation consisted of a rubber string with four steel electrodes (2-mm diameter and 40-

mm length), equally disposed (3 cm). The electric measurement sequence was based on both Wenner-Schlumberger and Dipole-Dipole quadrupoles. The sequence was repeated 3 times on each block to obtain the apparent resistivity along the same directions used for $V_P$-$V_S$ determinations and then averaged to obtain a representative electrical resistivity value of the sample. From the ratio between the measured electric potential difference, ΔV, and the injected current, I, the value of apparent resistivity, $R_{app}$, was obtained as follows:


$$R_{app} = k \, \frac{\Delta V}{I} \qquad (8)$$



where k is a geometric factor, depending on the geometry of the adopted quadrupole. For Wenner-Schlumberger and Dipole-Dipole sequences, k is respectively equal to:


$$k = 2 \cdot \pi \cdot a \qquad (9)$$

$$k = n \cdot (n + 1) \cdot (n + 2) \cdot \pi \cdot a \qquad (10)$$

where a is the electrode space of 3 cm, and n is equal to 1 because the electrodes are equally spaced.

The measurements were performed both in dry conditions and after saturation with tap water (electrical conductivity of 300 μS/cm). These repeated measurements, along with porosity and ultrasonic velocity measurements, were useful for better understanding rock matrix permeability and degree of fracturing/effective porosity. If the rock block exhibits high electrical resistivity in dry conditions and high resistivity in saturated conditions, it is plausible to consider it as

homogeneous, with resistivity close to the average of the minerals that compose the rock matrix. This holds true even with a high number of pores that are not interconnected. Conversely, a high resistivity value in dry conditions and low resistivity in saturated conditions indicate that the rock matrix contains a high number of connected pores/voids/fractures, that can be saturated with air in dry conditions and with the saturating fluid in wet conditions (water with electrical conductivity of 300 μS/cm in this study). While in dry conditions, the resistivity is hypothetically close to air resistivity

if the fracturing is very high, in saturated conditions, resistivity approaches the resistivity of the saturating fluid.

### 3.4 Permeability and UPV measurement at increasing confining pressure

Three samples, two CFB-FZ and one AFZ-HR, were selected and then tested in a pressure vessel at incremental effective pressures up to 80 MPa in order to measure permeability and UPV variation. Cylindrical shape samples with diameter equal to 40 mm and height of 80 mm were cored from the original cubic samples. Given the cm scale sample size, the

aim of these measurements was to determine the permeability of the matrix. During coring, fractures or obvious heterogeneities were avoided, focusing on capturing the pristine matrix. It is worth noting that the two CFB samples were sourced from different fault rock blocks characterized by a different amount of cementation. Measurements were performed at the Laboratoire de Géologie of Ecole Normale Superieure in Paris with a gas permeameter at different confining pressure steps (0.5, 2, 5, 10, 20, 40, 60, 80 MPa), in order to guarantee a gradual and progressive closure of the

voids space.

Permeability measurements were performed using the constant flow method under a fixed pressure gradient for permeability > $9.87*10^{-18}$ m$^2$ ($10^{-2}$ mD) and by using the pulse decay method (PDM), for permeability < $9.87*10^{-18}$ m$^2$ ($10^{-2}$ mD). The PDM method is based on the transient state induced in a porous media when the equilibrium state is perturbed by suddenly setting a pressure gradient in the sample. Typically, the upstream pressure ($P_u$) is increased while

downstream pressure ($P_d$) remains unchanged ($\Delta P_0 = P_u - P_d$). Upon perturbation, both up- and downstream pressures evolve with time ($\Delta P(t)$) following an inverse exponential trajectory (transient state) in their natural seeks for the restoration of an apparent equilibrium condition (i.e. steady state). The time (α) needed by the system to restore the apparent equilibrium condition depends on the dimensions of the sample, its properties, the volumes of the up- and downstream reservoirs ($V_u$, $V_d$), the physical characteristics of the fluid, and the applied pressure gradient ($\Delta P_0$).




$$\Delta P(t) = \Delta P_0 \frac{V_d}{V_u + V_d} e^{-\alpha t} \qquad (11)$$

Since the time decay α is function of permeability, of dynamic viscosity of the pore fluid at temperature of measurement, the geometric properties of the sample, and the compressive storage of the upstream ($C_u$) and downstream ($C_d$) reservoirs,

permeability values can be calculated via α for each step of increasing pressure. $C_u$ and $C_d$ are defined as the ratios of the change of fluid volume to the corresponding pore pressure variation (%). It is well known that gas can cause problems at low pore pressure due to the Klinkenberg effect, as follows:

$$k_{gas} = k \left(1 + \frac{b}{p_f}\right) \qquad (12)$$


where k denotes the ''true'' permeability, $p_f$, the mean pore pressure inside the sample and b the so-called Klinkenberg correction coefficient. Following Li et al. (2009) we assumed that b follows a power law with $b = 0.032k^{-0.43}$, with b in MPa and k in mD. This relation is closed to the power low found by Jones (1972) and Tanikawa and Shimamoto (2006) obtained on a large variety of rocks, resulting in $b = 0.048k^{-0.36}$, and $b = 0.053k^{-0.37}$, respectively. Using equation 12

and the power law, the Klinkeberg effect was taken in account and k was calculated for each relevant value of $p_f$ by solving equation (12) numerically (e.g., by means of the Newton's method).

For the UPV measurements, a 250-V pulse with a rise time of 1s was generated and transmitted to 2 Vp and Vs piezoelectric transducers placed at the top of each cylindrical sample. Each piezoceramic converted this electrical pulse into a mechanical vibration that propagated into the medium. Two receiving piezoceramic then converted the received

mechanical waveform into an electrical signal that was amplified at 40 dB with a sampling frequency of 50 MHz.

## 4   Experimental Results

### 4.1   Microstructural analyses

Figure 4 displays examples of the different analyzed textures of the block samples. Rocks collected from the various structural domains are classified into five groups based on the percentage of matrix versus clasts and the values of the

calculated $D_{0(pores)}$. Specifically, the main textures consist of: (i) matrix-supported carbonates (both host and fault-related rocks, Figure 4a and b), (ii) cement-supported carbonates (fault-related rocks, Figure 4c), (iii) clast-supported carbonates (host rocks, Figure 4d), (iv) survivor grain-supported carbonates (fault-related rocks, Figure 4e), and (v) fragmented carbonates (host rocks, Figure 4f). Table 1 summarizes the main results obtained from digital image analyses performed on the rock slabs. Specifically, it reports the percentage of matrix versus clast and the value of $D_{0(grains)}$.




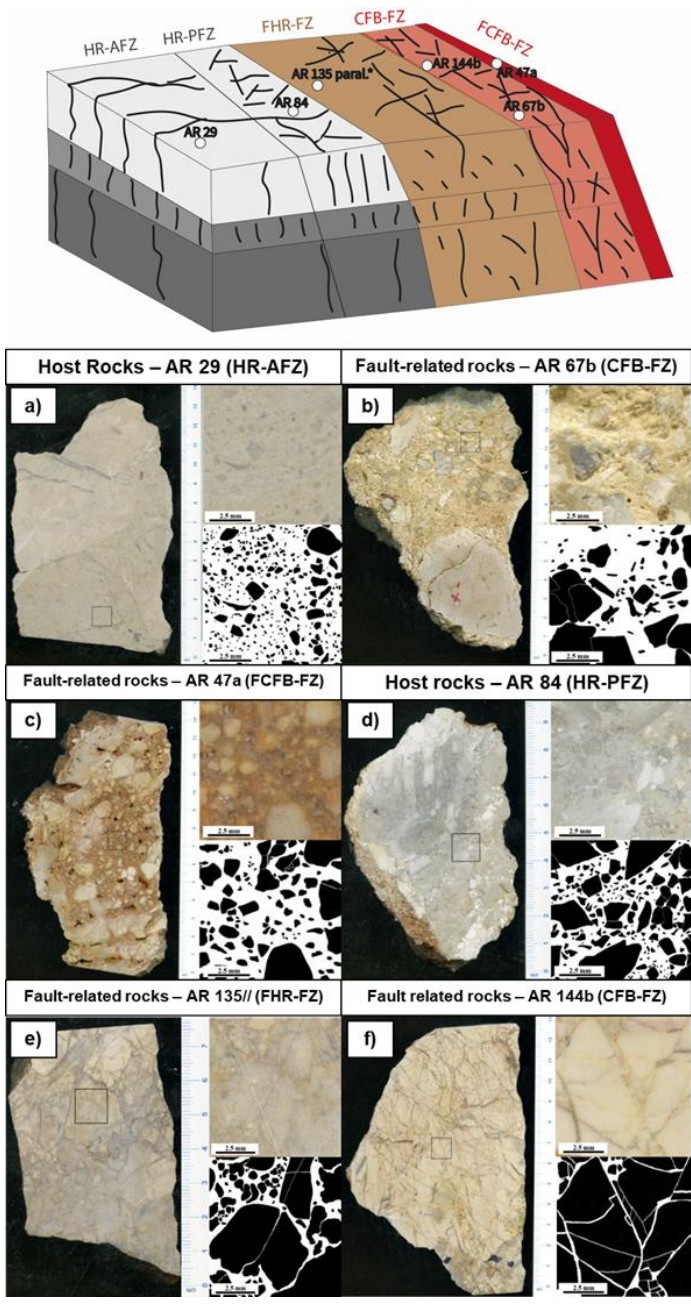

**Figure 4: Schematic cross-section of the footwall damage zone of the studied carbonate fault zones, in which the main structural domains are highlighted. Names of representative samples are reported for the single domains. The prevalent textures of the carbonate samples are the following: a) matrix-supported carbonates (host rocks), b) matrix-supported carbonates (fault-related rocks), c) cement-supported carbonates (fault-related rocks), d) clast-supported carbonates (host rocks), e) survivor grain-supported carbonates (fault-related rocks), and f) fragmented carbonates (fault-related rocks). The black square represents the 1 cm² selected image for subsequent digital analysis. Upper right: 1 cm² image for subsequent digital analysis.**




**Table 1. Summary of image analyses performed on both host rocks and fault rocks.**

|  | Slab | % of clasts | % of Matrix | $D_{0(grains)}$ |
|---|---|---|---|---|
| HR-AFZ | AR 1P | 21 | 79 | 1.59 |
|  | AR 5 | 95 | 5 | 1.96 |
|  | AR 29 | 25 | 75 | 1.62 |
|  | AR 44c | 52 | 48 | 1.8 |
| HR-PFZ | AR 84 | 57 | 43 | 1.84 |
|  | AR 84bis | 92 | 8 | 1.97 |
| FHR-FZ | AR 43paral.# | 94 | 6 | 1.96 |
|  | AR 43perp.# | 94 | 6 | 1.96 |
|  | AR 71 | 80 | 20 | 1.92 |
|  | AR 103 | 83 | 17 | 1.93 |
|  | AR 117 | 59 | 41 | 1.83 |
| CFB-FZ | AR 121 | 86 | 14 | 1.93 |
|  | AR 122 | 67 | 33 | 1.86 |
|  | AR 28# | 77 | 23 | 1.92 |
|  | AR 59 | 41 | 59 | 1.74 |
|  | AR 67 b | 46 | 54 | 1.78 |
|  | AR 75 b | 52 | 48 | 1.82 |
|  | AR 761 | 67 | 33 | 1.87 |
|  | AR 762 | 51 | 49 | 1.8 |
|  | AR 78 | 43 | 47 | 1.76 |
|  | AR 81# | 60 | 40 | 1.85 |
|  | AR 109# | 26 | 74 | 1.64 |
|  | AR 132# | 46 | 54 | 1.76 |
|  | AR 133 | 57 | 43 | 1.83 |
|  | AR 135paral.# | 75 | 25 | 1.91 |
|  | AR 135perp.# | 80 | 20 | 1.94 |
|  | AR 140 | 66 | 34 | 1.86 |
|  | AR 142# | 76 | 24 | 1.92 |
|  | AR 144a | 78 | 22 | 1.91 |
|  | AR 144b | 87 | 13 | 1.94 |
| FCFB-FZ | AR 17 a1 | 88 | 12 | 1.94 |
|  | AR 17 a2 | 64 | 36 | 1.85 |
|  | AR 17 b | 67 | 33 | 1.86 |
|  | AR 47 a | 44 | 56 | 1.76 |
|  | AR 47 b | 70 | 30 | 1.88 |
|  | AR 53 | 41 | 59 | 1.74 |
|  | AR 61 | 53 | 47 | 1.8 |
|  | AR 61penta a | 52 | 48 | 1.8 |
|  | AR 61penta b | 34 | 66 | 1.69 |
|  | AR 65# | 36 | 64 | 1.71 |
|  | AR 69 | 49 | 51 | 1.78 |
|  | AR 83 | 90 | 10 | 1.96 |
|  | AR 88# | 50 | 50 | 1.79 |
|  | AR 131paral.# | 65 | 35 | 1.86 |
|  | AR 131perp.# | 43 | 57 | 1.75 |
|  | AR 1411# | 38 | 62 | 1.72 |
|  | AR 1412# | 93 | 7 | 1.96 |
|  | AR 145 | 32 | 68 | 1.7 |


2D digital image analyses of selected thin sections impregnated in epoxy resins are performed to quantify 2D porosity, $D_{0(pores)}$, and other pore characteristics such as aspect ratio and width. The 6 host rock samples collected either away from (HR-AFZ) or in proximity to fault zones (HR-PFZ) consist of 4 Senonian limestone and sedimentary breccia, and of 2

Vigla limestone. All six samples include stylolite, microfractures, and calcite veins. Pores very often align along microfractures, but also form vugs due to not selective dissolution of both clasts and matrix-cements (Figure 5a and 5b). These samples show a mean 2D porosity $\leq 0.7\%$, and $D_{0(pores)} \leq 1.15$.



The 5 FHR-FZ samples collected from mesoscale fault zones consist of 1 Senonian limestone (2 rock slabs) collected from a N110E striking Fault Zone, FZ, 1 Senonian limestone collected from a N90E striking fault zone, 1 Senonian

sedimentary breccias collected from a N110E FZ, and 1 Vigla limestone collected from a N150E FZ. All five samples include microfractures and veins. Pores mainly align along the former structural elements. Both small and large vugs due to not-selective dissolution of both clasts and matrix/cements are present (Figure 5c). Most of the samples show a mean 2D porosity from ~ 4.12% to ~ 7.71%, and $D_{0(pores)}$ range from ~ 1.24 to ~ 1.56. Differently, the fractured carbonate packstone is characterized by average values 2D porosity~0.21%, and a mean $D_{0(pores)}$ equal to 1.4.

The 19 CFB-FZ samples from fault zones consist of 2 samples collected from a N10E FZ, 2 samples from a N20E FZ, 7 samples from a N45E FZ, 4 samples from a N100E FZ, 3 samples from a N110E FZ, and 1 sample from a N130E FZ. The cement/matrix-supported fault rock textures include vugs due to not-selective dissolution of both clasts and matrix/cements, and pores that often align around the edges of survivor grains (Figure 5d). Most of the samples show a mean 2D porosity ≤ ~ 3%, with some vlaues≥5; $D_{0(pores)}$ ranges from ~ 0.6 to ~ 1.62.

The 18 FCFB-FZ samples from fault zones consist of 1 sample collected from a N10E FZ, 2 samples from a N20E FZ, 3 samples from a N45E FZ, 1 sample from a N100E FZ, 9 samples from a N110E FZ, and 2 samples from a N130E FZ. The samples include numerous microfractures and veins. Pores are mainly aligned along the former structural elements. Vugs due to not-selective dissolution of both survivor grains and matrix/cements are present (Figure 5e). The 2D porosity range from ~ 0.31% to 8.52&, and $D0_{(pores)}$ from ~ 0.91 to ~ 1.76.






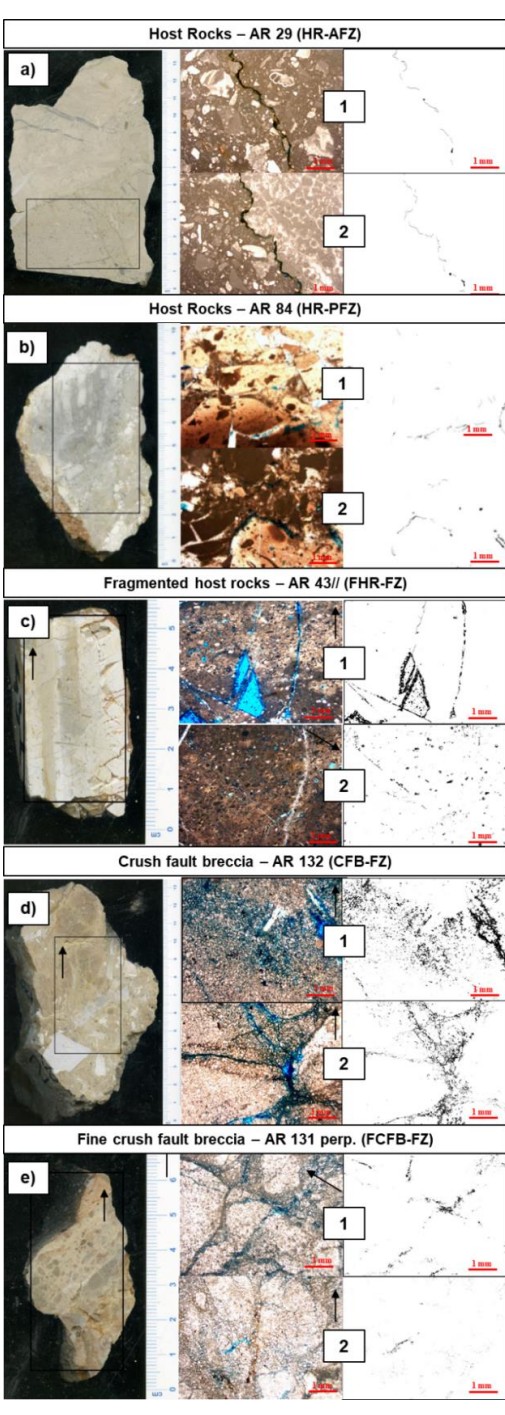

**Figure 5: The main pore types documented in a) host rocks, b) fragmented host rock, c) crush fault breccia, d) fine crush fault breccia. The black square on the rock slab represents the thin section area. In the digital images, pores are in black, clasts, matrix and cement in white.**



Data obtained via digital image analysis of the pore space characteristics are computed in order to obtain the average, median, and 1σ standard deviation values of both pore aspect ratio (AR) and pore aperture (width, in σm) for host rocks (HR-AFZ, HR-PFZ, FHR-FZ, Table 2), and for fault rocks (CFB-FZ, FCFB-FZ, Table 3). Data are reported for all pores, and for those with AR ≥ 2. The mean values of median AR and width are in bold for both tables. The latter values correspond to the minor axis of the ellipse automatically inscribied within each individual pore.


**Table 2: Summary of the main pore characteristics (Aspect Ratio, AR, and Width, in σm) computed for the samples collected either away from or in proximity to meso-scale fault zones. The aforementioned characteristics are reported for all of the pores depicted in the whole images, and for pores with AR > 2. Numbers in bold represent the mean values computed for single thin sections.**

**HR-AFZ & HR-PFZ**

| | # | ASPECT RATIO | | | | | | WIDTH (μm) | | | | | |
|---|---|---|---|---|---|---|---|---|---|---|---|---|---|
| | | whole image | | | AR > 2 | | | whole image | | | AR > 2 | | |
| | | AR average | AR median | AR st dev | AR average | AR median | AR st dev | WIDTH average | WIDTH median | WIDTH st dev | WIDTH average | WIDTH median | WIDTH st dev |
| SENONIAN WACKESTONE | AR 1 P | | **1,52** | | | **2,18** | | | **1,13** | | | **2,67** | |
| | 1 | 1,66 | 1,52 | 0,74 | 2,43 | 2,18 | 0,58 | 9,7 | 1,13 | 37,96 | 18,9 | 2,67 | 61,89 |
| | 2 | | | | | | | | | | | | |
| SENONIAN MUDSTONE | AR 5 | | **2,985** | | | **3,9** | | | **8,935** | | | **11,97** | |
| | 1 | 2,5 | 1,68 | 1,59 | 3,54 | 3,08 | 1,57 | 6,9 | 2,67 | 7,31 | 11,6 | 8,94 | 3,95 |
| | 2 | 4,88 | 4,29 | 3,24 | 5,35 | 4,72 | 3,2 | 8,6 | 15,2 | 11,5 | 16,5 | 15 | 9,42 |
| SENONIAN BRECCIA | AR 29 | | **2,715** | | | **4,54** | | | **9,12** | | | **13,95** | |
| | 1 | 4,6 | 2,8 | 4,65 | 6,56 | 5,33 | 4,89 | 21,6 | 11,8 | 25,5 | 20.04 | 15,7 | 18,2 |
| | 2 | 3,55 | 2,63 | 3,14 | 5,04 | 3,75 | 3,19 | 11,8 | 6,44 | 13,8 | 15 | 12,2 | 12,4 |
| SENONIAN BRECCIA | AR 44 c | | **1,285** | | | **2,49** | | | **3,245** | | | **3,355** | |
| | 1 | 1,81 | 1,57 | 0,92 | 2,68 | 2,59 | 0,89 | 10,5 | 3,82 | 19,5 | 11 | 4,04 | 20,4 |
| | 2 | 1,51 | 1 | 0,9 | 2,7 | 2,39 | 0,98 | 4,28 | 2,67 | 5,51 | 5,43 | 2,67 | 7,25 |
| VIGLA PACKSTONE | AR 84 | | **1,51** | | | **2,535** | | | **2,67** | | | **3,765** | |
| | 1 | 1,96 | 1,47 | 2,09 | 3,14 | 2,42 | 2,94 | 6,12 | 2,67 | 8,91 | 7,43 | 3,75 | 10,4 |
| | 2 | 1,96 | 1,55 | 1,41 | 3,09 | 2,65 | 1,68 | 7,63 | 2,67 | 11,7 | 8,47 | 3,78 | 12,2 |
| VIGLA MUDSTONE | AR 84 Bis | | **16,52** | | | **16,52** | | | **39,7** | | | **39,7** | |
| | 1 | 5,37 | 4,68 | 2,32 | 5,37 | 4,68 | 2,32 | 51,2 | 56,5 | 18,3 | 51,2 | 56,5 | 18,3 |
| | 2 | 28,36 | 28,36 | 20,77 | 28,36 | 28,36 | 20,77 | 22,9 | 22,9 | 14,8 | 22,9 | 22,9 | 14,8 |

**FHR-FZ**

| | # | ASPECT RATIO | | | | | | WIDTH (μm) | | | | | |
|---|---|---|---|---|---|---|---|---|---|---|---|---|---|
| | | whole image | | | AR > 2 | | | whole image | | | AR > 2 | | |
| | | AR average | AR median | AR st dev | AR average | AR median | AR st dev | WIDTH average | WIDTH median | WIDTH st dev | WIDTH average | WIDTH median | WIDTH st dev |
| VIGLA FR. MUDSTONE | AR 71 | | **1,23** | | | **2,48** | | | **2,67** | | | **3,075** | |
| | 1 | 1,63 | 1 | 1,43 | 2,77 | 2,31 | 2,09 | 4,64 | 2,67 | 28,6 | 7,69 | 2,67 | 50,4 |
| | 2 | 1,99 | 1,46 | 2,54 | 3,63 | 2,65 | 3,9 | 7,85 | 2,67 | 25,9 | 12,7 | 3,48 | 36,3 |
| SENONIAN FR. MUDSTONE | AR 103 | | **1,45** | | | **2,555** | | | **2,06** | | | **2,47** | |
| | 1 | 2,16 | 1,45 | 2,94 | 4,02 | 2,54 | 4,45 | 8,31 | 1,45 | 2,94 | 12,6 | 3,75 | 20,4 |
| | 2 | 1,62 | 1,45 | 0,74 | 2,54 | 2,57 | 0,55 | 10,4 | 2,67 | 60,7 | 15,6 | 1,19 | 35,44 |
| SENONIAN | AR 117 | | **3,015** | | | **3,545** | | | **9,585** | | | **9,585** | |



| | | | | | | | | | | | | | |
|---|---|---|---|---|---|---|---|---|---|---|---|---|---|
| BRECCIA | 1 | 1,72 | 1,59 | 0,77 | 2,48 | 2,65 | 0,41 | 21,4 | 2,67 | 55,8 | 28,9 | 2,67 | 73,6 |
| | 2 | 5,12 | 4,44 | 2,78 | 5,12 | 4,44 | 2,78 | 18,8 | 16,5 | 6,62 | 18,8 | 16,5 | 6,62 |


**Table 3: Summary of the main pore characteristics (Aspect Ratio, AR, and Width, in μm) computed for the fault breccia and fine crush fault breccia and fault microbreccia samples collected from meso-scale fault zones. The aforementioned characteristics are reported for all of the pores depicted in the whole images, and for pores with AR > 2. Numbers in bold represent the mean values computed for single thin sections.**


### CFB- FZ

| # | ASPECT RATIO | | | | | | WIDTH (μm) | | | | | |
|---|---|---|---|---|---|---|---|---|---|---|---|---|
| | whole image | | | AR > 2 | | | whole image | | | AR > 2 | | |
| | AR average | AR median | AR st dev | AR average | AR median | AR st dev | WIDTH average | WIDTH median | WIDTH st dev | WIDTH average | WIDTH median | WIDTH st dev |
| AR 12 V | | **1,23** | | | **2,075** | | | **2,67** | | | **2,67** | |
| 1 | 1,52 | 1 | 0,66 | 2,36 | 2 | 0,5 | 3,59 | 2,67 | 2,25 | 3,47 | 2,67 | 2,84 |
| 2 | 1,63 | 1,46 | 0,71 | 2,42 | 2,15 | 0,59 | 3,95 | 2,67 | 2,73 | 3,67 | 2,67 | 1,88 |
| AR 59 V | | **1,23** | | | **2,475** | | | **2,67** | | | **3,095** | |
| 1 | 1,63 | 1 | 0,91 | 2,65 | 2,48 | 0,89 | 4,54 | 2,67 | 8,2 | 5,25 | 2,67 | 3,67 |
| 2 | 1,7 | 1,46 | 0,9 | 2,7 | 2,47 | 0,87 | 15,3 | 2,67 | 43 | 14,3 | 3,52 | 37,4 |
| AR 67b V | | **1,43** | | | **2,635** | | | **2,67** | | | **2,74** | |
| 1 | 1,89 | 1,57 | 1,15 | 2,82 | 2,65 | 1,19 | 5,21 | 2,67 | 9,94 | 5,48 | 2,81 | 11,9 |
| 2 | 1,7 | 1,29 | 0,95 | 2,66 | 2,62 | 0,88 | 4,34 | 2,67 | 12,1 | 4,36 | 2,67 | 11,6 |
| AR 75b V | | **1,535** | | | **2,65** | | | **2,67** | | | **2,905** | |
| 1 | 1,9 | 1,46 | 1,27 | 2,95 | 2,65 | 1,3 | 4,49 | 2,67 | 11,1 | 4,95 | 2,69 | 8,82 |
| 2 | 1,97 | 1,61 | 1,26 | 2,97 | 2,65 | 1,31 | 5,37 | 2,67 | 18,9 | 5,15 | 3,12 | 7,03 |
| AR 76_1 V | | **1,27** | | | **2,595** | | | **3,175** | | | **3,4** | |
| 1 | 1,81 | 1,54 | 1,02 | 2,76 | 2,54 | 1,07 | 8,51 | 3,68 | 27,8 | 11,5 | 3,78 | 43,1 |
| 2 | 1,68 | 1 | 1,14 | 2,91 | 2,65 | 1,35 | 5,77 | 2,67 | 19,7 | 9,43 | 3,02 | 34,3 |
| AR 76_2 V | | **1,355** | | | **2,63** | | | **2,67** | | | **2,74** | |
| 1 | 1,71 | 1,21 | 0,95 | 2,66 | 2,61 | 0,88 | 4,22 | 2,67 | 7,88 | 4,47 | 2,67 | 8,71 |
| 2 | 1,83 | 1,5 | 1,04 | 2,79 | 2,65 | 1 | 6,45 | 2,67 | 18,9 | 7,08 | 2,81 | 25,9 |
| AR 78 V | | **1,315** | | | **2,47** | | | **2,685** | | | **3,15** | |
| 1 | 1,61 | 1 | 0,96 | 2,65 | 2,39 | 1,1 | 5,91 | 2,67 | 15,3 | 6,28 | 2,67 | 15,5 |
| 2 | 1,93 | 1,63 | 1,19 | 2,88 | 2,55 | 1,26 | 8,1 | 2,7 | 25 | 7,32 | 3,63 | 15,4 |
| AR 133 V | | **1,515** | | | **3,01** | | | **4,28** | | | **5,225** | |
| 1 | 3,96 | 1,74 | 4,7 | 7,47 | 3,4 | 5,43 | 25,1 | 5,89 | 55,15 | 21,1 | 7,78 | 33,66 |
| 2 | 1,7 | 1,29 | 0,95 | 2,66 | 2,62 | 0,88 | 4,34 | 2,67 | 12,1 | 4,36 | 2,67 | 11,6 |
| AR 140 V | | **1,575** | | | **2,63** | | | **3,635** | | | **3,795** | |
| 1 | 1,77 | 1,55 | 0,9 | 2,7 | 2,61 | 0,86 | 8,28 | 3,82 | 21,6 | 11,3 | 3,78 | 33,8 |
| 2 | 2,03 | 1,6 | 1,52 | 3,12 | 2,65 | 1,78 | 7,54 | 3,45 | 17,5 | 7,86 | 3,81 | 17,2 |
| AR 144a V | | **1,745** | | | **3,29** | | | **3,605** | | | **3,73** | |
| 1 | 4,27 | 2,49 | 3,76 | 5,63 | 3,93 | 3,79 | 10,39 | 4,54 | 19,45 | 6,35 | 4,65 | 3,97 |
| 2 | 2,03 | 1 | 2,6 | 3,6 | 2,65 | 3,76 | 5,78 | 2,67 | 14,5 | 8,43 | 2,81 | 20,4 |
| AR 144b V | | **1** | | | **2,65** | | | **2,67** | | | **2,31** | |
| 1 | 1,26 | 1 | 0,58 | 2,55 | 2,65 | 0,49 | 3,23 | 2,67 | 3,84 | 2,63 | 2,31 | 0,6 |
| 2 | | | | | | | | | | | | |

### CFB-FZ

| # | ASPECT RATIO | | | | | | WIDTH (μm) | | | | | |
|---|---|---|---|---|---|---|---|---|---|---|---|---|
| | whole image | | | AR > 2 | | | whole image | | | AR > 2 | | |
| | AR average | AR median | AR st dev | AR average | AR median | AR st dev | WIDTH average | WIDTH median | WIDTH st dev | WIDTH average | WIDTH median | WIDTH st dev |
| AR 17a | | **1,425** | | | **2,665** | | | **2,67** | | | **2,74** | |
| 1 | 1,98 | 1,85 | 1,13 | 2,84 | 2,65 | 1,07 | 4,67 | 2,67 | 2,3 | 4,6 | 2,81 | 4,89 |
| 2 | 2,27 | 1 | 2,95 | 4,6 | 2,68 | 4,1 | 4,43 | 2,67 | 5,6 | 7,46 | 2,67 | 8,71 |
| AR 17b | | **1,23** | | | **2,25** | | | **2,67** | | | **2,74** | |
| 1 | 1,63 | 1 | 1,27 | 2,65 | 2,32 | 1,77 | 3,89 | 2,67 | 5,,56 | 4,56 | 2,67 | 8,01 |
| 2 | 1,67 | 1,46 | 1,01 | 2,62 | 2,18 | 1,24 | 5,23 | 2,67 | 3,1 | 6,31 | 2,81 | 10,8 |
| AR 47a | | **1,465** | | | **2,65** | | | **2,67** | | | **2,74** | |
| 1 | 1,75 | 1,47 | 0,91 | 2,65 | 2,65 | 0,81 | 4,93 | 2,67 | 13,38 | 4,82 | 2,67 | 13,1 |

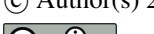



| | | | | | | | | | | | | |
|---|---|---|---|---|---|---|---|---|---|---|---|---|
| 2 | 1,76 | 1,46 | 0,94 | 2,75 | 2,65 | 0,75 | 11,8 | 2,67 | 27,7 | 17,9 | 2,81 | 37,8 |
| **AR 47b** | | **1,5** | | | **2,54** | | | **2,67** | | | **2,67** | |
| 1 | 1,91 | 2 | 1,04 | 2,75 | 2,65 | 0,8 | 10,7 | 2,67 | 58,5 | 16,2 | 2,67 | 80,6 |
| 2 | 1,59 | 1 | 0,96 | 2,7 | 2,43 | 1,11 | 15,6 | 2,67 | 92 | 18,3 | 2,67 | 78,1 |
| **AR 53** | | **1,465** | | | **2,7** | | | **2,67** | | | **4,13** | |
| 1 | 1,99 | 1,67 | 1,14 | 2,96 | 2,7 | 1,01 | 12,1 | 2,67 | 43 | 20,9 | 3,07 | 62 |
| 2 | 2,06 | 1,26 | 2,14 | 3,73 | 2,7 | 2,92 | 19 | 2,67 | 55,5 | 24,9 | 5,19 | 66,3 |
| **AR 61** | | **1,505** | | | **2,47** | | | **3,245** | | | **3,47** | |
| 1 | 1,73 | 1,46 | 1,14 | 2,71 | 2,45 | 1,45 | 7,53 | 2,67 | 18,4 | 8,26 | 3,02 | 22,7 |
| 2 | 1,93 | 1,55 | 1,67 | 3 | 2,49 | 2,31 | 13 | 3,82 | 32,2 | 10,9 | 3,92 | 18,8 |
| **AR 61penta_a** | | **1,465** | | | **2,38** | | | **2,67** | | | **2,81** | |
| 1 | 1,72 | 1,46 | 1,21 | 2,74 | 2,35 | 1,57 | 5,65 | 2,67 | 11,8 | 6,18 | 2,81 | 12,4 |
| 2 | 1,72 | 1,47 | 0,84 | 2,58 | 2,41 | 0,76 | 5,67 | 2,67 | 12 | 6,14 | 2,81 | 13,5 |
| **AR 61penta_b** | | **1,5** | | | **2,39** | | | **2,67** | | | **3,2** | |
| 1 | 1,73 | 1,46 | 1,22 | 2,68 | 2,32 | 1,59 | 6,41 | 2,67 | 26,7 | 7,36 | 2,81 | 27,4 |
| 2 | 1,75 | 1,54 | 0,87 | 2,6 | 2,46 | 0,83 | 6,58 | 2,67 | 17,5 | 6,06 | 3,59 | 12,2 |
| **AR 69** | | **1,235** | | | **2,35** | | | **2,67** | | | **2,74** | |
| 1 | 1,55 | 1 | 0,82 | 2,52 | 2,3 | 0,84 | 4,28 | 2,67 | 9,81 | 5,5 | 2,67 | 16,2 |
| 2 | 1,74 | 1,47 | 1,21 | 2,67 | 2,4 | 1,59 | 6,18 | 2,67 | 16,8 | 7,35 | 2,81 | 25,5 |
| **AR 145** | | **1,6** | | | **2,305** | | | **10,41** | | | **4,4** | |
| 1 | 1,67 | 1,46 | 0,74 | 2,56 | 2,3 | 0,69 | 18,6 | 3,82 | 65,4 | 33,8 | 2,84 | 111 |
| 2 | 1,81 | 1,74 | 0,51 | 2,31 | 2,31 | 0,34 | 61,8 | 17 | 108 | 7,71 | 5,96 | 5,97 |

## 4.2 Petrophysical characterization at room pressure and temperature

Figure 6 and 7 show the density and porosity values measured for each rock sample. The average normalized value obtained for each rock block group (HR-AFZ to FCFB-FZ) is summarized in Table 4. Normalization of the average values
is achieved by considering the maximum and minimum values among all rock blocks in both dry and wet conditions. In general, density of both host rocks (HR-AFZ and HR-PFZ) and fractured host rock (FHR-FZ) varies between 2500 and 2900 kg/m³ and porosity is on average lower than 3%. A few samples from the Vigla and Senonian formations show porosity values up to 6%. Differently, CFB-FZ and FCFB-FZ samples show lower values of density, ranging from 2200 to 2900 kg/m³. These samples show also higher porosity with some values up to 25%. Overall, both CFB-FZ and FCFB-
FZ samples are quite heterogeneous in terms of petrophysical characteristics depending upon distance from the main slip surfaces and pervasiveness of reddish carbonate cements. In general, density decreases approaching the main slip surfaces (dashed trend lines in Figure 6), whereas porosity increases (dashed trend line in Figure 7).



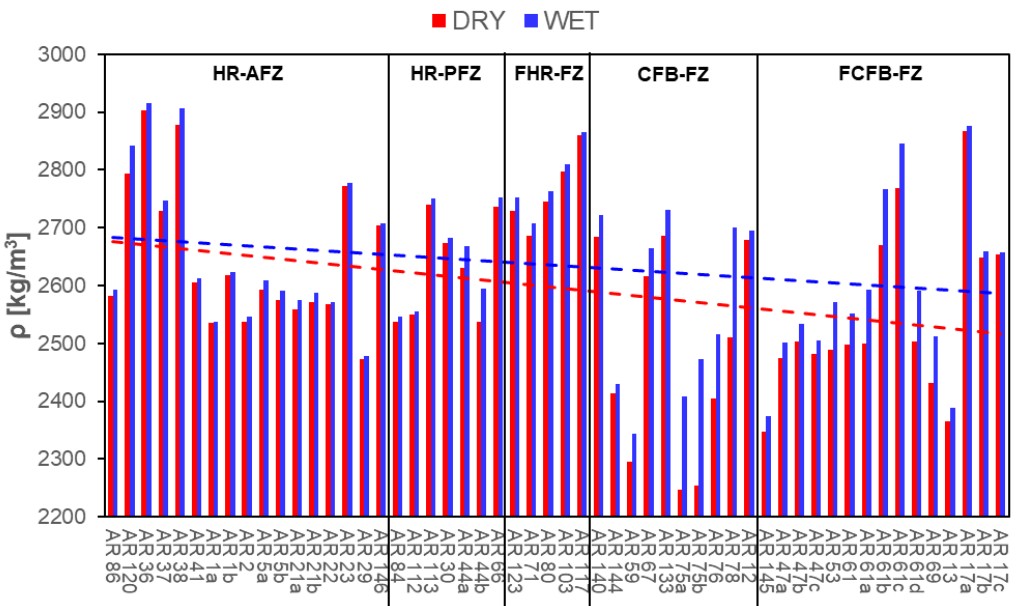

**Figure 6: Dry (red bars) and saturated (blue bars) density values for all the rock samples collected in the surrounding of the fault zone.**

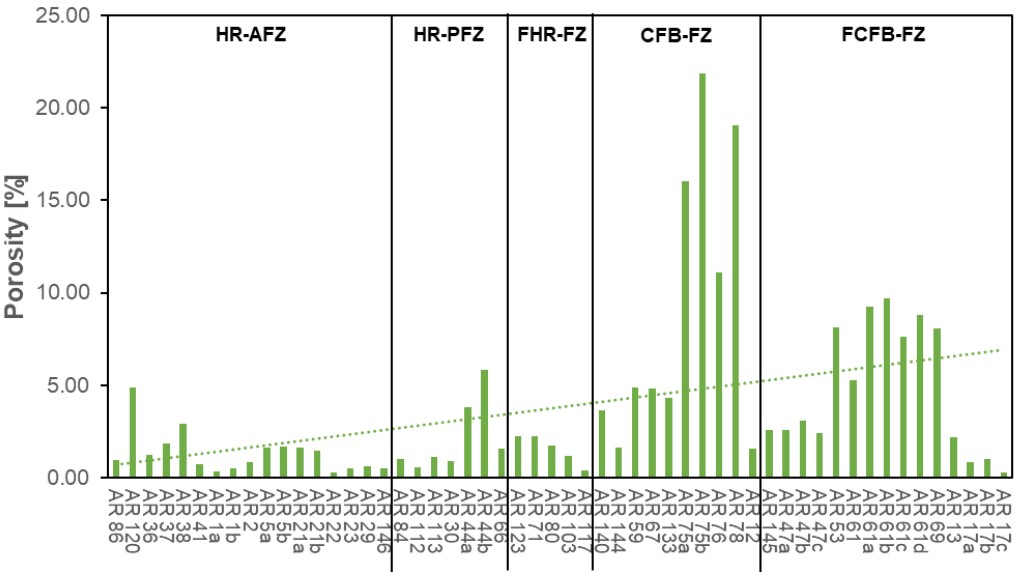

**Figure 7: Porosity values for all the rock samples collected in the surrounding of the fault zone.**

UPV measurements are performed both in dry and in saturated conditions along three orthogonal directions. The obtained values are averaged to provide a mean value for each sample. Similarly, the wave velocities of each samples are averaged



to obtain a mean normalized value representative of the rock block group (Table 4), employing the method previously described. Figure 8 and 9 show the $V_P$ and $V_S$ values. $V_P$ (dry conditions) varies between 6400 m/s and 2600 m/s. $V_S$ (dry conditions) ranges between 3600 and 1200 m/s. . VP (wet conditions) varies between 6700 m/s and 2700 m/s, meanwhile for VS in wet conditions, there are no appreciable differences compared to the values in dry conditions. Considering the

aforementioned ultrasonic data, $V_{P\,WET}$ is always greater than the $V_{P\,DRY}$. This discrepancy is higher for samples collected close to the main slip surfaces, mirroring the increase in porosity shown in Figure 7. Conversely, the $V_S$ values remain almost unchanged (cf. Figure 9).

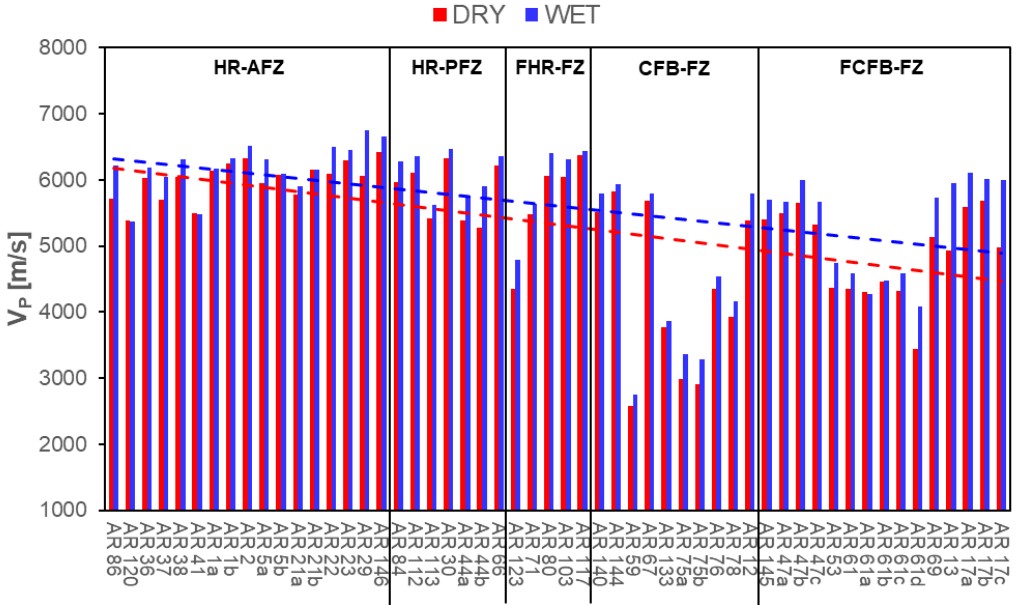

**Figure 8: $V_P$ values in dry (red bars) and saturated (blue bars) conditions evaluated as the average of the measurements along**
**the three directions for each rock sample collected in the surrounding of the fault zone.**





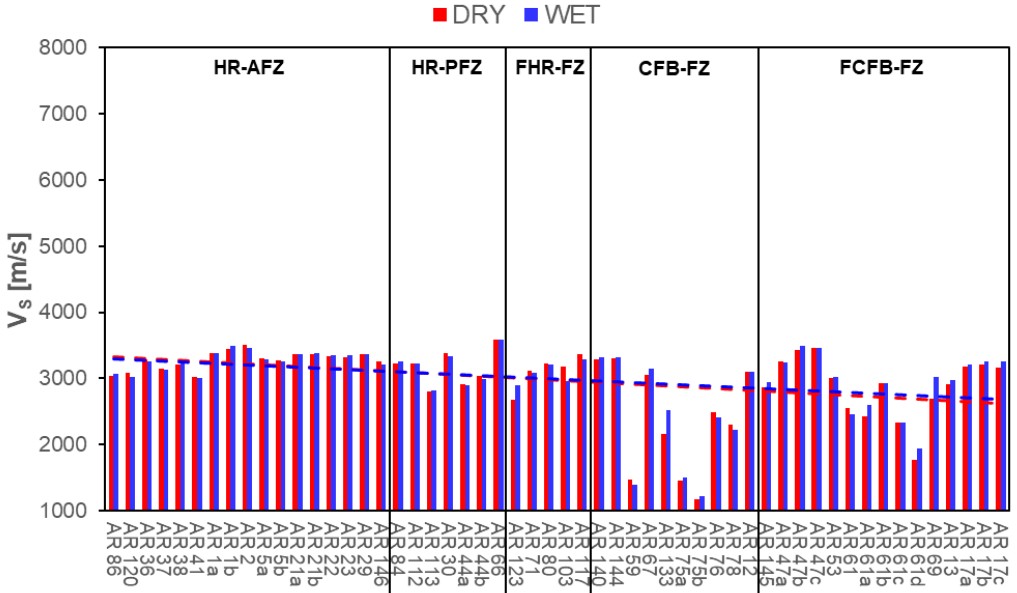

**Figure 9: V_S values in dry (red bars) and saturated (blue bars) conditions evaluated as the average of the measurements along the three directions for each rock sample collected in the surrounding of the fault zone.**

Figure 10 shows $V_{Pdry}$ -$V_{Sdry}$ computed for each rock sample variations and reported as coefficient of anisotropy, K. For the sake of simplicity and readability, only the results obtained in dry conditions are presented in the figure, as the results in wet conditions are almost identical. K is estimated as following:

$$K = \frac{V_{i_{max}}}{V_{i_{min}}} \qquad (12)$$


with $V_i$ being ultrasonic velocity (both P and S) in dry or wet conditions. The lower limit of K is 1, indicating no differences between the wave velocities in the three directions, hence an isotropic rock behavior. Higher values reflect anisotropy, which can be transversal isotropic (textural anisotropy compatible to a preferential orientation of grains or planar anistrotropy compatible to bedding interfaces, fractures, slip surfaces) if there is a marked velocity in one direction 395 while the other two directions exhibit the same values, or due to higher complexity given to the overlap of different micromechanisms acting on the rock texture and thus undefined if the three direction values are different. Figure 10 provides general information on the isotropic or anisotropic behavior of the rock. However, if considered as standalone observation, it cannot reveal the nature of the anisotropy measured.

Indeed, there is not a clear directivity in the trend of the coefficient of anisotropy. Overall, the highest K values are 400 observed in the carbonate fault rocks relative to the host rocks. Such a result is consistent with a textural anisotropy profoundly affected by tectonic fractures and slip surfaces. In detail, K is up to 1.2 in HR-AFZ, and up to 1.4 in HR-PFZ. For the latter rocks, the highest computed values are for blocks AR 113 (Figure 10) and AR 44, which correspond to Senonian microbreccia samples collected in proximity to N110E FZs respectively crosscutting the topmost and bottom portions of the Senonian carbonates. The microstructural analysis suggests that these anisotropies are due to mesoscale



fractures dissecting the host rock blocks. FHR-FZ samples are also characterized by K values up to 1.2, confirming their

close similarities with the previous ones. Regarding both CFB-FZ and FCFB-FZ, K values are respectively up to 1.8 (AR

59) and 2.3 (AR 69). The AR 59 block derives from a N110E FZ crosscutting the bottom Senonian succession, whereas

the AR69 block derives from a small N130E FZ adjacent to the previous one. The  microstructural analysis shows that

these anisotropies can be associated to mesoscale fractures and/or sheared fractures that sub-parallel the main slip

surfaces.

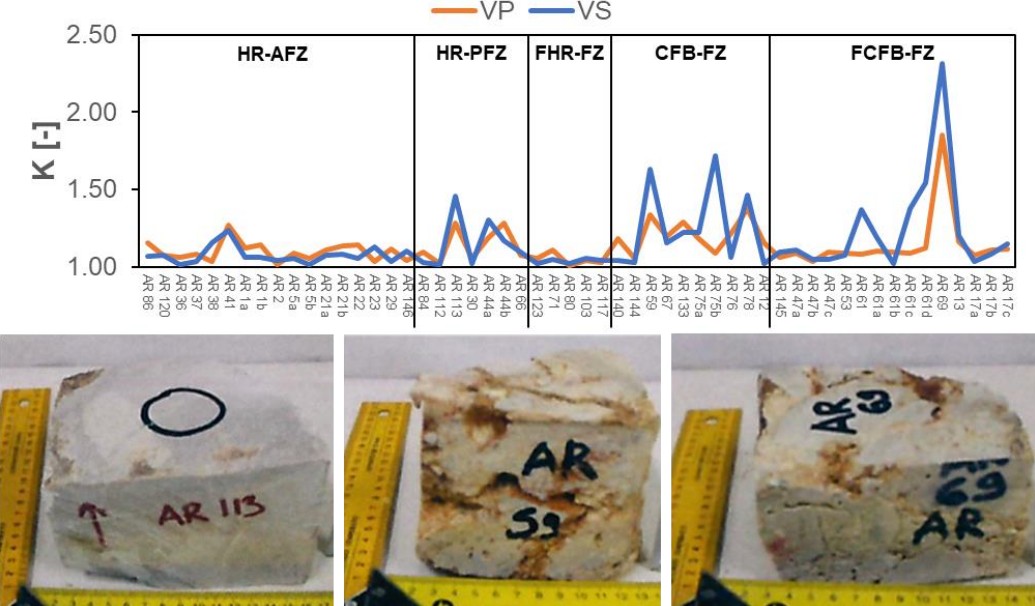

**Figure 10: Trend of the coefficient of anisotropy, K, evaluated considering seismic properties for all blocks, along with photographs of the three representative rock blocks with the highest K value.**

The dynamic Young's and shear modulus, as well as Poisson's ratio, are evaluated from the seismic values obtained for

the single structural domains and shown as average normalised values (Table 4). It is noteworthy that the dynamic moduli

computed for the host rock blocks (HR-AFZ, HR-PFZ, and FHR-FZ) are higher than those obtained for the fault rock

blocks (CFB-FB and FCFB-FZ). These variations match the rock properties higlighted by the ultrasonic pulse velocity

and electrical resistivity tetsts, and show an inverse correlation with the effect of fracturing (porosity). These values will

be therefore useful for the indirect estimation of the evolution of crack density at the meso- and site-scale by using the

theory proposed by (Nasseri et al., 2007), as discussed in Section 5.

The Electrical Resistivity (ER) values measured for the carbonate rock blocks are summarized in Table 4. There, the

average normalized resistivity values obtained for both dry, $R_{DRY,N}$, and saturated conditions, $R_{WET,N}$, are reported. Data

show that while resistivity in dry condition remains relatively constant with a slight increase approaching the fault zone,

the resistivity in saturated conditions shows a marked decrease, in good agreement with an high degree of fracturing.





Table 4. Average normalized physical values for each rock group.

|  | HR-AFZ | HR-PFZ | FHR-FZ | CFB-FZ | FCFB-FZ |
|---|---|---|---|---|---|
| Normalized Dry Density [-] | 0.91 | 0.90 | 0.95 | 0.85 | 0.87 |
| dev.st | 0.11 | 0.09 | 0.16 | 0.11 | 0.09 |
| Normalized Wet Density [-] | 0.91 | 0.91 | 0.95 | 0.88 | 0.89 |
| dev.st | 0.11 | 0.09 | 0.16 | 0.11 | 0.10 |
| Normalized Porosity [-] | 0.06 | 0.10 | 0.07 | 0.41 | 0.22 |
| dev.st | 0.01 | 0.01 | 0.01 | 0.05 | 0.02 |
| Normalized Dry $V_P$ [-] | 0.89 | 0.86 | 0.84 | 0.64 | 0.72 |
| dev.st | 0.04 | 0.05 | 0.02 | 0.05 | 0.04 |
| Normalized Wet $V_P$ [-] | 0.92 | 0.90 | 0.88 | 0.67 | 0.79 |
| dev.st | 0.04 | 0.04 | 0.03 | 0.05 | 0.04 |
| Normalized Dry $V_S$ [-] | 0.91 | 0.88 | 0.87 | 0.66 | 0.80 |
| dev.st | 0.03 | 0.05 | 0.02 | 0.05 | 0.06 |
| Normalized Wet $V_S$ [-] | 0.91 | 0.88 | 0.86 | 0.67 | 0.82 |
| dev.st | 0.03 | 0.05 | 0.05 | 0.05 | 0.05 |
| Normalized Dry E [-] | 0.82 | 0.76 | 0.77 | 0.45 | 0.59 |
| dev.st | 0.01 | 0.01 | 0.01 | 0.00 | 0.01 |
| Normalized Dry G [-] | 0.81 | 0.75 | 0.76 | 0.45 | 0.61 |
| dev.st | 0.02 | 0.02 | 0.02 | 0.01 | 0.02 |
| Normalized Dry μ [-] | 0.68 | 0.69 | 0.66 | 0.67 | 0.54 |


## 4.2 Permeability and UPV measurement at confining pressure

Figure 11 shows permeability (k), corrected for the Klinkenberg effect, and UPV determination under different target confining pressure during gas permeameter tests. A good agreement between UPV measurements performed at room pressure (Figure 8 and 9) and those at the first confining step (0.5 MPa) is found for each tested sample. As a general

pattern, permeability decreases with increasing $V_P$ and $V_S$ at increasing confining pressures. We note that the permeability of sample AR78 (Figure 11b) is higher than all others due to lack of any cement within the collected CFB-FZ specimen, which is also characterized by higher values of porosity and crack density (cf. Figure 7). Also, sample AR12 (CFB-FZ, Figure 11c) shows similar values of both permeability and velocity to sample AR41 (HR-AFZ, Figure 11a) due to cementation processes.






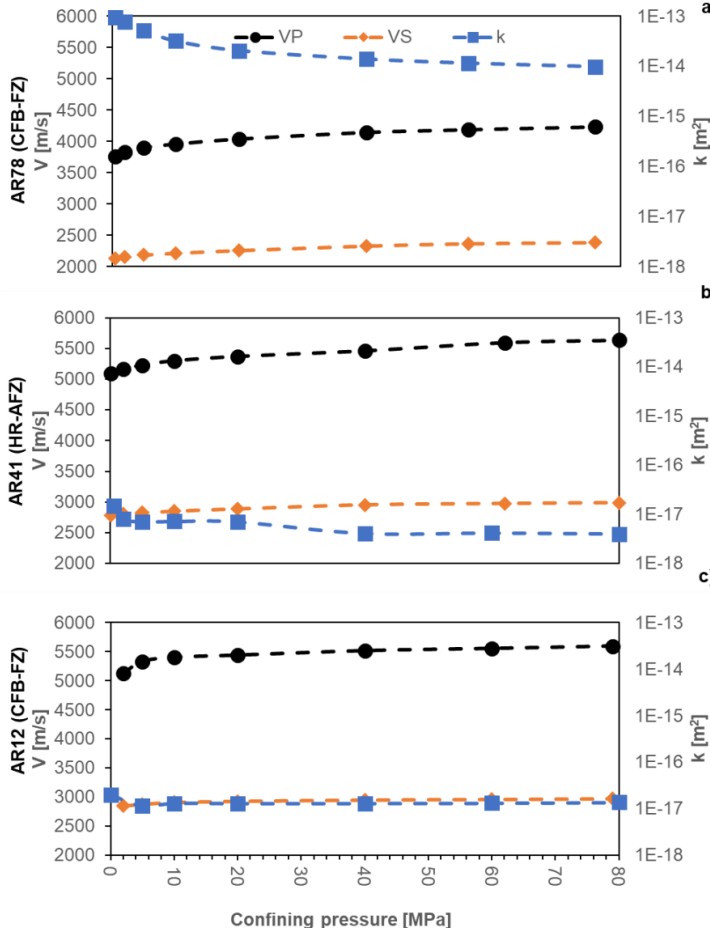

**Figure 11: Evolution of permeability (blue symbols), V$_P$ (black symbols) and V$_S$ (orange symbols) as a function of confining pressure during gas permeameter tests on sample a) AR78 (CFB-FZ), b) AR41 (HR-AFZ) and c) AR12 (CFB-FZ).**

### 5 Modelling elastic properties and crack density

Fluid flow is controlled by the connected network of cracks. Consequently, defining the key features describing the crack network geometry such as crack density, size, aspect ratio, and alignment is essential in any attempt of permeability prediction. Gueguen and Dienes (1989) showed that permeability of cracked rocks can be well represented by that of an array of penny-shaped cracks embedded in an impermeable host matrix. Using the isotropic formulation of their percolation model, the bulk rock permeability, k, can be expressed as follows:


$$k = \frac{2}{15} \cdot f \cdot w^2 \cdot \zeta \cdot \rho \qquad (13)$$

where $f$ is the percolation factor, $w$ is the crack average width, $\zeta$ is the average aspect ratio, and $\rho$ is the crack density.



*f*, can vary between 0 and 1. When *f* is equal to 0 it means that all cracks are isolated and unconnected; consequently, rocks can be considered as intact with a very low degree of fracturing. Conversely, when *f* corresponds to the unity, all cracks are connected into the network. Guéguen et al. (1997) showed that the percolation threshold (defined as the minimum crack density required to establish a complete path for a fluid to flow) for microscopic permeability is reached at values of crack density equals to 0.14. Differently, the percolation threshold for macroscopic fractures is reached at values of crack density close to 1. The percolation factor may be approximated by the connectivity factor, *f'*, as defined by (Gueguen and Dienes, 1989):

$$f' = \frac{9}{4}\left(\frac{\pi^2}{4}\rho - \frac{1}{3}\right) \qquad (14)$$

Since changes in dynamic elastic properties are correlated to the amount of damage within rocks, it is possible to invert the elastic wave velocity field into a non-dimensional crack density, $\rho$, as defined by the following equation:

$$\rho = \frac{1}{V}\sum_{i=1}^{N} c_i^3 \qquad (15)$$

where $c_i$ is the radius of the i$^{th}$ crack, and N the total number of cracks embedded in the representative elementary volume V.

The Effective Medium Theory (EMT) introduces the concept of "no stress interaction approximation" by neglecting, as a simple assumption, the stress interactions among cracks. In this case, the effective elastic moduli of a cracked solid can be exactly and rigorously calculated by solely considering crack orientation and distribution (Bristow 1960, Kachanov 1993) . The theory invokes that for cracks randomly oriented and distributed (perfect isotropic configuration), stress interactions are partially geometrically compensated so that the effective Young's modulus, E*, of a dry rock can be obtained as follows:

$$\frac{E_0}{E^*} = 1 + \frac{16\left(1-\nu_0^2\right)\left(1-\frac{3\nu_0}{10}\right)}{9\left(1+\frac{\nu_0}{2}\right)}\rho \qquad (16)$$

where $E_0$ and $\nu_0$ are the petrophysical Young's modulus and Poisson ratio of the un-cracked material, respectively, and E* is the Young's modulus calculated by using the ultrasonic pulse measurements.

The crack aspect ratios, $\zeta$, can be also evaluated from results of digital image analysis, as described in Section 3.2, or it may be calculated by simply applying the following equation:

$$\zeta = \frac{<w>}{<2c>} \qquad (17)$$

Recalling that for penny-shaped cracks with constant aspect ratio distribution, the total crack porosity, n, is also equal to:

$$n = \pi w c^2 \qquad (18)$$



the aspect ratios, $\zeta$, can be therefore expressed as a function of both porosity, n, and crack density, $\zeta$, as shown by the following equation:

$$\zeta = \frac{n}{2\pi\rho} \qquad (19)$$

495

Two workflows (WF) are followed to estimate permeability (k) from seismic measurements. WF1 applies the Gueguen and Dienes (1989) and EMT theories; the values of aspect ratio and crack width are assessed on the basis of the results of seismic measurements. WF2 assumes a constant crack aperture (width) of 1μm, which is typical of a crack network geometry. For both WF crack density of the analyzed rock blocks is evaluated by inverting equation 16, using E* from seismic measurements, and assuming representative petrophysical values of $E_0$ and $v_0$ for matrix-supported carbonates (host rocks, fault rocks), cement-supported carbonates (fault rocks), clast-supported carbonates (host rocks), and survivor grain-supported carbonates (fault rocks) on the basis of a bibliographic review of similar lithologies (Mavko et al. 2009, Table 5). The crack aspect ratio is estimated by using eq. 19. For WF1, once crack density ($\zeta$) was calculated, the crack average width is obtained from equations 17 and 18. Conversely, following WF2, the crack average width ($w$) is considered constant and equal to 1 μm. Finally permeability (k) is evaluated by using equation 13; since the percolation factor ($f$) is extremely complex to measure, and k is sensitive to its variation, an interval analysis by considering lower and upper limits in the interval of $f$ variation (f equal to 0 and 1) and an intermediate value provided by equation 14 was performed. Moreover, a WF3 employing the crack density ($\zeta$) values obtained from digital image analyses is applied.

510 **Table 5. Summary of the representative petrophysical $E_0$ and $v_0$ values used for the crack density estimation.**

| Lithology | E0 [GPa] | $v$0[-] |
|---|---|---|
| cement-supported fault breccia | 90 | 0.25 |
| matrix-supported host rocks | 83 | 0.29 |
| matrix-supported fault rocks | 80 | 0.30 |
| survivor grain-supported fault rock | 87 | 0.31 |
| fragmented host rock | 82 | 0.30 |

Figure 12 shows the variation of *k*, by considering the three aforementioned workflows, and *f* of 0.01 (blue line) and 1 (orange line). *k* obtained by considering the percolation factor equals to the connectivity factor ($f'$) are also reported (grey line). *k* experimentally obtained on three samples (cf. Ch. 4.3) at the lowest confining pressure are also shown with black symbols for comparison. Overall the host rock permeability (HR-AFZ, HR-PFZ, FHR-FZ) varies between $10^{-17}$ and $10^{-14}$ m$^2$ (0.01 and 10 mD), with a few samples exhibiting *k* values falling outside this range. Differently, CFB-FZ and FCFB-FZ samples shows higher permeability values up to $10^{-13}$ m$^2$ (100 mD).

For what it concerns the comparison with k values from laboratory tests, the k inferred from seismic measurements (WF1) best reproduces the experimental results, especially when percolation factor is evaluated with equation 14 (grey line in Figure 12a). WF2 and WF3 do not reproduce the higher permeability values (up to $10^{-13}$ m$^2$ - 100 mD). Considering WF2 (Figure 12b), the calculated *k* values are generally lower, ranging between $10^{-18}$ and $10^{-15}$ m$^2$ (0.001 and 1 mD), with a slight increase for the fault rock samples. A constant crack aperture not only determines lower values of calculated *k* , but



it also tends to smooth out the differences. The results obtained by following WF3 (Figure 12c) show a very slight increase of fault rock $k$ with respect to the previous case.



**Figure 12: Trend of permeability (orange, grey and blue lines) and porosity (green bars) values evaluated for each rock block by following: a) protocol 1, b) protocol 2, and c) protocol 3. The three reported permeability trends are the results of different percolation factor values: 0.01 for the blue lines, 1 for the orange lines, and estimated by using equation 15 for grey lines. The black diamonds identify the rock block permeability measured with gas permeameter.**



## 6    Discussion

In this section, the results of laboratory analyses performed both at the micro- and mesoscale were compared to understand the nature and geometrical characteristics of the pores, in order to establish poro-perm relations for the lithologies of the Araxos Promontory. In particular, microscale analyses allowed the definition of the main mechanisms that rock samples
underwent during deformation phases. Mesoscale correlation between physical and estimated mechanical parameters and porosity is provided to assess the properties of single structural domains, and validate the hypotheses made on permeability estimation from indirect measurements, as discussed previously.

### 6.1    Deformation mechanisms

In order to document the scale-invariant process of deformation documented at the Araxos Promontory, and the
progressive comminution of the carbonate fault breccia, we plot $D_{0(grains)}$ vs. % of matrix values obtained after digital images (Figure 16). Data show a very good fitting ($R^2 = 0.99$) and no significant change of its slope. Any slope change would have produced lower $R^2$ values, related to a variation of the deformation mechanisms during fault rock comminution (Ferraro et al., 2018). This correlation can be associated to the diffused bulk crushing (cf. Figures 4 and 5) associated with high-angle faulting (Bourli et al., 2019a; Smeraglia et al., 2023). It is also highlighted that the cementation
might have coated together the pre-existing survivor grains, obliterating therefore the possible effects of chipping and shear fracturing in the study fault rocks.

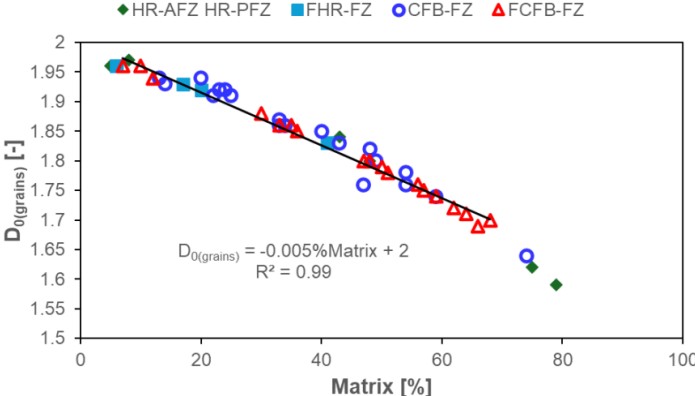

**Figure 13: Plots of box-counting Dimension, $D_{0(grain)}$, vs. % of matrix for all host rock and fault-related rock samples. Data**
**points related to the single measurements are reported in the log-log diagram according to the single lithologies.**

### 6.2    Pore properties

In order to assess the pore types, we plot the average $V_P$ in dry conditions versus porosity values with respect to the Hashin Strickman upper and lover bounds (HS+ and HS-, respctively), the time-average equation (Wyllie et al., 1958, 1956), and the empirical best fit line proposed by (Anselmetti and Eberli, 2001, 1997, 1993) for carbonates (Figure 14).
The HS+, HS-, and time-average equation are calculated for a monimineralic, calcite-rich rock. The former parameters represent the narrowest possible bounds for elastic moduli calculated for an isotropic material by only knowing the volume fractions of the constituents (Mavko et al., 2009), whereas the time-average equation relates the $V_P$ of an isotropic,



fluid-saturated, consolidated rock to its porosity assuming that the total travel time can be approximated as the volume-
weighted average of travel times through the individual rock constituents. The best-fit curve predicts the P-wave velocity
of carbonates at any given porosity. Values of $V_P$ higher than the best fit are due to presence of molds, which form by
selective dissolution on the carbonates not affecting their elastic frame and thus enhancing porosity but not permeability,
values lower that the best fit are due to vugs and microfractures.

According to the results of Figure 14, data are consistent with the vast majority of the blocks being affected by vug
porosity due to a not-selective dissolution of the carbonates and microfractures. Some of the host rock data points lie
along the Anselmetti and Eberli's (2001) best fit line, whereas all fault rock data points show presence of
vugs/microfractures.

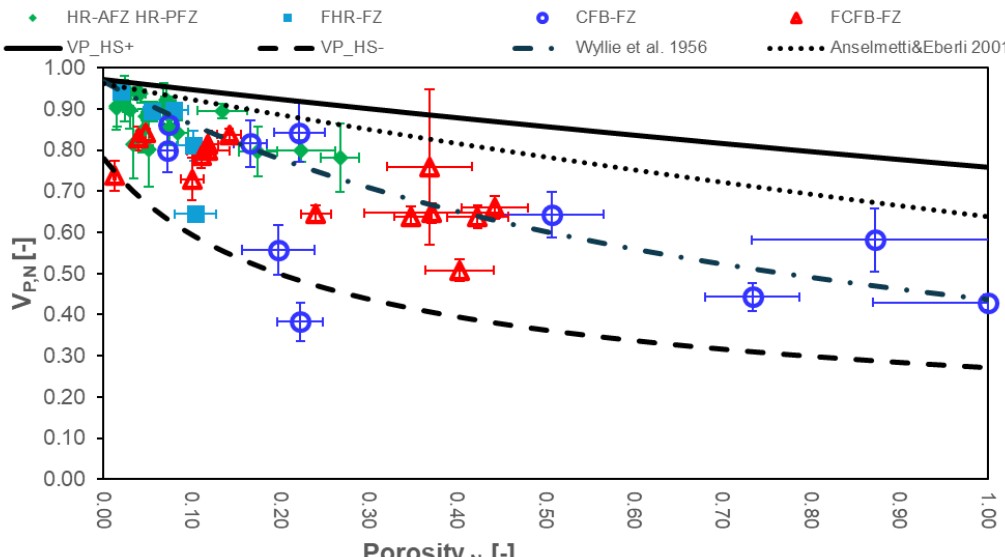

**Figure 14: Pore type of carbonate host rocks and fault rocks deciphered after the comparison between normalized $V_{P,DRY}$ and**
**porosity values. The Hashin Strickman upper (H+), and lower bounds (H-), the Wyllie's time average equation (Wyllie et al.**
**1956), and the linear best fit equation proposed by Anselmetti and Eberli (2001) for carbonates of Italy and Bahamas are also**
**reported.**

The pore volume distribution in carbonates can be fractal (Wu et al., 2019, and references therein), and statistically
described by a power-law function (Mandelbrot, 1985). If the spatial pattern of the pores is also fractal, the plot of the
box size versus the filling frequency is power-law, and corresponds to the angular coefficient of the best fit line ($D_{0(pores)}$,
Falconer, 2003; Ferraro et al., 2018). Considering $D_{0(pores)}$ vs. 2D$\varphi$ (Figure 15), they show a positive correlation. $D_{0(pores)}$
increases with higher 2D $\varphi$, meaning that pores are more uniformly distributed in higher porosity carbonates, which
correspond to the fault rocks. In other words, we observe that rock comminution due to brecciation caused a more
homogeneous distribution of the microfractures throughout the investigated 1 cm$^2$ areas. We also note that the very low
value of the fitting coefficient ($R^2 = 0.27$, Figure 15) is due to outliers likely associated to the effect of not-selective
dissolution of the carbonates, which affected both host rocks and fault rocks.



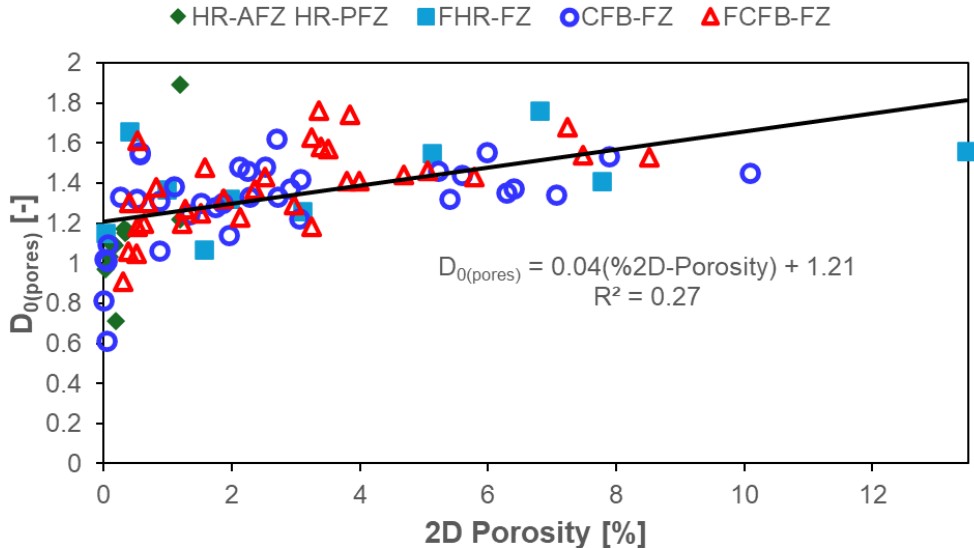

**Figure 15: D0(pores) vs. % of 2D porosity for all rocks.**

### 6.3 Physico-mechanical parameters vs porosity

Figure 16 illustrates the variation of the normalized petrophysical and mechanical parameters ($V_P/V_{S,DRY,N}$, $E_{DRY,N}$ and $v_{DRY,N}$), estimated from UPV measurements, along with the electrical properties, $R_{DRY}/R_{WET}$ with respect to the normalized porosity for each studied rock group. We only report the trend of parameters estimated in dry conditions, since the trend in saturated conditions fully reproduces the same behaviour.

    Outcomes strengthen our findings based on microstructural analyses. In fact, the $V_P/V_{S,DRY,N}$ trend, which provides

information on the stiffness and, to some extent, the poro-elasticity of the rock, is constant over the range of normalized porosity, with a slight increase for both host rocks (HR-AFZ and HR-PFZ) and CFB-FZ samples. This trend reflects two opposite physical and mechanical behaviours: 1) host rocks hold stiff properties, with high Young's modulus (Figure 16c) and a low $R_{DRY}/R_{WET}$ , indicating a low difference between electrical resistivity in dry and saturated conditions. This suggests also a higher homogeneity and isotropy of the pores distribution, as also evidenced by the microstructural

analysis and the lowest values of anisotropy coefficient (Figure 17). On the contrary, fault tocks such CFB-FZ and FCFB-FZ samples exhibit very low mechanical properties (Figure 16c) and high deformability due to higher values of Poisson's ratio (Figure 16b), as well as a high electrical $R_{DRY}/R_{WET}$, due to the high degree of fracturing (Figure 17). In terms of estimated permeability (Figure 12), regardless of the chosen workflow, CFB-FZ samples have higher values. Interesting is the behaviour of, which from a mechanical and permeability point of view, are similar to CFB-FZ samples. In terms of

porosity, the distribution of FCFB-FZ samples appears to be more uniform with respect to CFB-FZs, which show preferred directionality (refer to Figure 5).



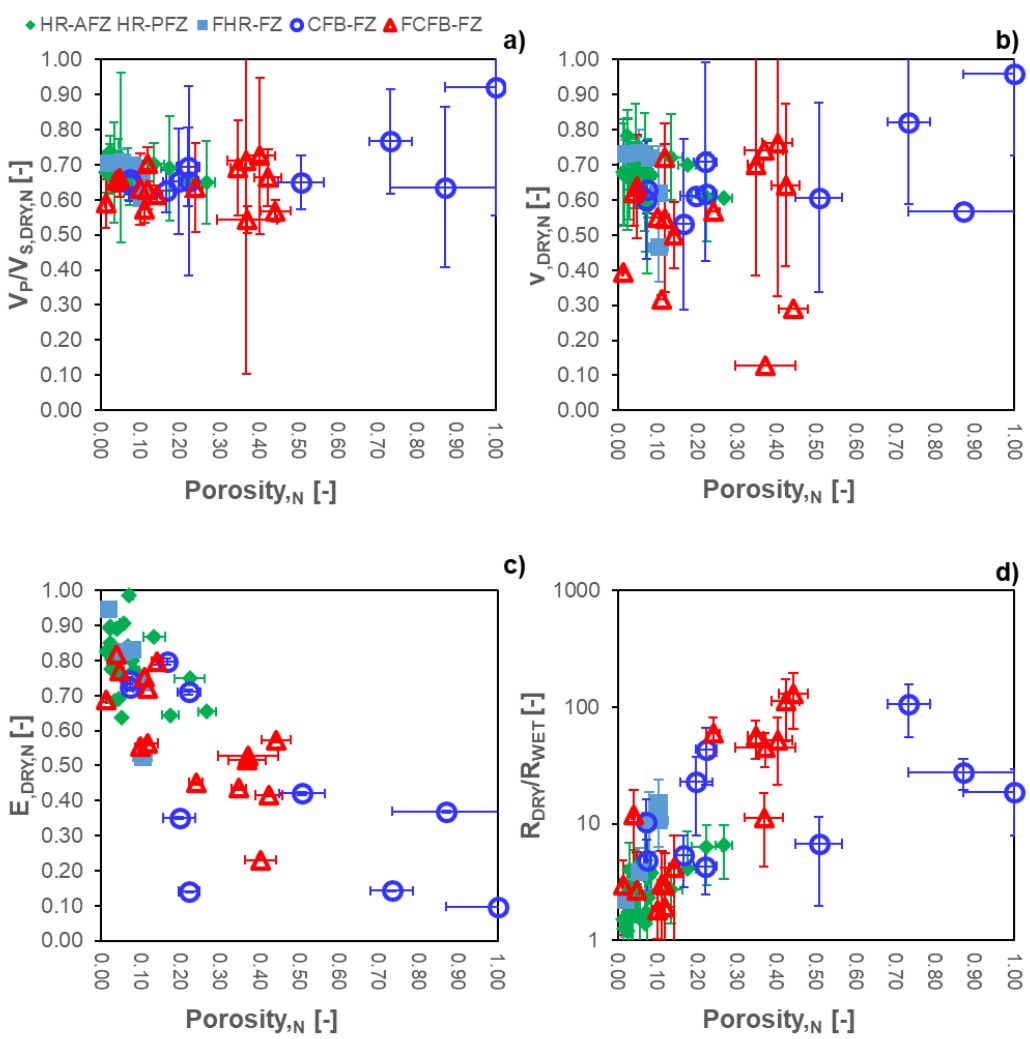

**Figure 16: Relationship between normalized a) $V_P/V_S$ in dry conditions, b) normalized Poisson's ratio, v, c) normalized Young's modulus, E, and d) ratio between electrical resistivity in dry condition and in saturated conditions, $R_{DRY}/R_{SAT}$, and porosity for the studied rock blocks.**



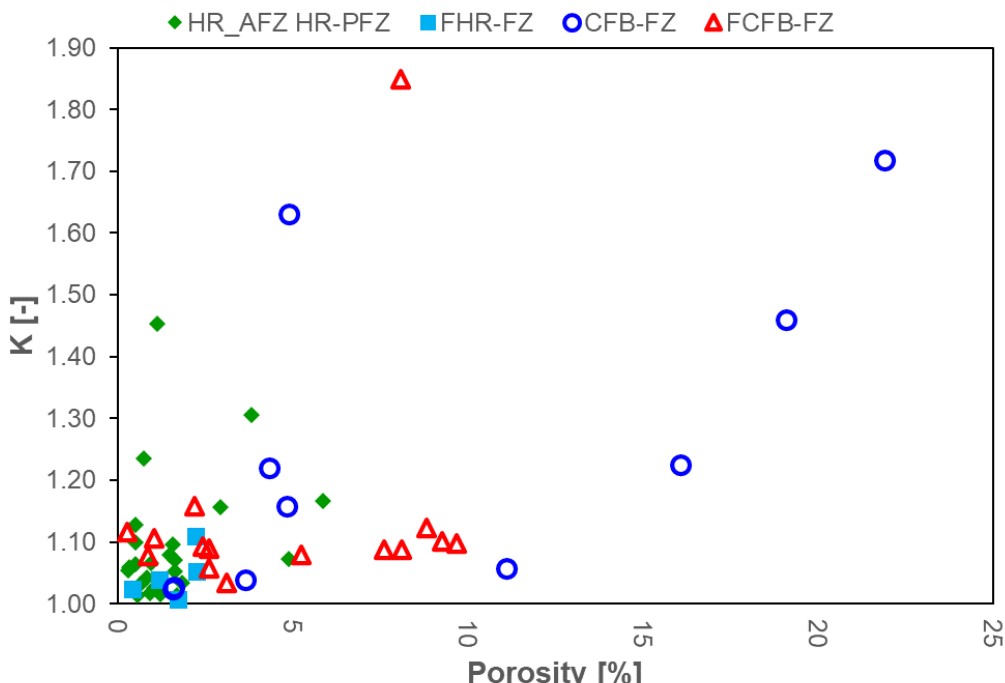

**Figure 17: Relationship between the coefficient of anisotropy, K, and porosity for the studied rock blocks.**

### 6.4 Poro-perm relations

The poro-perm relations (Figure 18) are assessed for each workflow described in Section 5 by considering the percolation factor, f defined with equation 14. The grey continuous lines represent the variability domain defined by considering the upper (f = 1) and lower (f = 0.01) edges of the range of variability of percolation factor f. As shown in Figure 18, the carbonate host rocks (HR-AFZ, HR-PFZ, FHR-FZ - green and cyan markers) do not show any significant poro-perm relation. In fact, the calculated k values exhibit a large permeability variation, up to 4 orders of magnitude, and do not vary proportionally with porosity.

The aforementioned lack of relation is interpreted as due to presence of both stiff, sub-rounded pores and of vugs/microfractures in the carbonate host rocks (Agosta et al, 2007; Ferraro et al., 2020). Accordingly, the host rock data points show a wide dispersion in the plot of Figure 18. On the contrary, the calculated k values of fault rocks log-linearly increase with porosity. Although fault rocks (CFB-FZ and FCFB-FZ) show a very large permeability variation, up to 5 order of magnitude, the log-linear relation is interpreted as due to a connected pore network (Ehrenberg et al., 2006), which mainly includes vugs/microfractures (refer to Section 6.2).

The poro-perm relations are in good agreement, regardless of the WF chosen for their determination. There is also a good correspondence with the laboratory tests, although not completely exhaustive due to the limited number, supporting the hypothesis that expedite methodologies, such as those via geophysical methods supported by micro-structural analyses, are reliable for evaluating permeability.

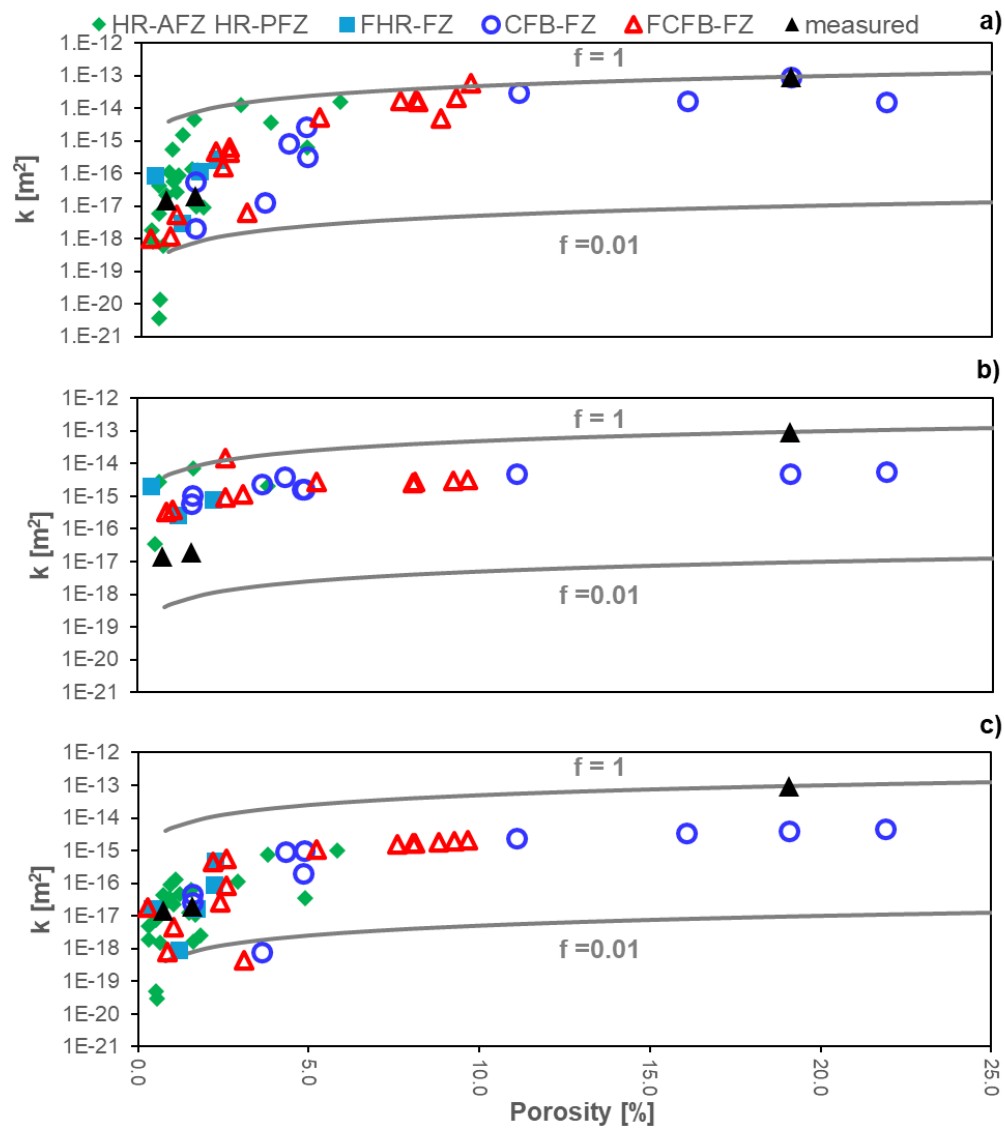

**Figure 18: Poro-perm relations for both host rocks (HR-AFZ and HF-PFZ) and fault rocks (FHR-FZ, CFB-FZ, FCFB-FZFR) considering the connectivity factor, f' (equation 14), and adopting a) workflow 1 (EMT theory), b) workflow 2 (w= 1 μm) and**
**c) workflow 3 (2D image analysis).**

## 7   Conclusions

We present the results of a petrophysical characterization of Mesozoic carbonate rocks collected from outcrops of the Araxos Promontory, northwest Greece, aiming to define the poro-perm relation of both host rocks and fault rocks. Rock
samples consist of Senonian and Vigla formations made of carbonate mudstones, wackestones, packstones and



sedimentary breccias, and of fault breccias collected from high-angle extensional and strike-slip fault zones. Compared to the host rocks, results show that the fault breccias exhibit a wider range of density, up to 5-10 times greater porosity, and lower ultrasonic velocities. A slight textural anisotropy is documented in the carbonate host rocks, whereas a more evident anisotropy characterizes the fault rocks mirroring the increase of porosity. Regardless of lithology, the carbonate
host rocks include small vugs, wherea the fault breccia also include microfractures. Selected samples were also tested in pressure vessels with confining pressure up to 80MPa, showing that transport properties along microcracks in fault breccias can significantly increase with increasing depth.

To compute the poro-perm relations, three different workflows (WF) were employed to estimate permeability: i) from the Effective Medium Theory, by considering the results of ultrasonic measurements; ii) a constant crack aperture of 1
μm and iii) crack density values from 2D image analysis. The results highlighted that the carbonate host rocks did not show a clear poro-perm trend due to the presence of stiff, sub-rounded pores and small vugs. The fault breccia showed a linearly increase of permeability with porosity due to a connected pore network built by microfractures. Regarding the employed WF, a good correspondence between theoretical calculations and gas permeability measurements was observed. These findings support the reliability of using expedited analyses, such as geophysical methodologies supported by
microstructural analyses for the determination of permeability in fault zones. In particular, this methodology is useful for detailed outcrops characterization or for a preliminary analysis of retrieved rock samples to improve geological modeling with particular application to engineering geology where understanding fluid dynamics is fundamental, such as in petroleum or geothermal reservoirs.

**Competing interests**

The contact author has declared that none of the authors has any competing interests.

**Acknowledgements**

Scientific discussion with C. Turrini and E. Panza is acknowledged. We thank S. Grimaldi and C. Manniello for the help provided during fieldwork activities. Financial support from Hellenic Petroleum is acknowledged. Fieldwork activities and laboratory analyses were also supported by the Reservoir Characterization Project (www.rechproject.com).

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
