# Peer review of "Poro-perm relations of Mesozoic carbonates and fault breccia, Araxos Promontory, NW Greece"

_EGUsphere, 2024_

## Author Comment (AC1)

RC1: This manuscript investigates the petrophysical characteristics of rock samples collected in NW Greece, with special attention to their role in geological formations Combining few structural analyses with laboratory measurements, the results of two-dimensional images and petrophysical analyses are synthesized to investigate various carbonate lithologies, calculate permeability values for different types of rocks, and compare them with experimental measurements. The results are discussed on the basis of direct observation of the connected pore space, and a scheme for the petrophysical analysis of carbonate rocks is proposed. The experimental part of the manuscript is well designed and supports the conclusions better, however, there are still some issues.

RE: Thanks for the positive assessment of our manuscript.

RC1: The first is the abstract section, which should begin with an introduction of the field to which this research belongs and highlight its importance in order to draw out the focus of this research. The current version is more oriented to what one did, which got some results, which will shed the interest of the general public and greatly reduce the impact of the manuscript. It is recommended that the authors that section be reorganized.

RE: Ok, we aimed to present the results found in the abstract as well, but we can reorganize it according to the reviewer suggestions.

RC1: Sample collection and testing is an important part. The authors should consider to further describe the sampling strategy, I didn't see enough justification for the sampling choices in the manuscript, whether the samples are representative or not is an important thing.

RE: The representativeness of the study samples is a key point to consider, for this reason we provide a synoptical 3D representation of the main structural domains assessed for the study fault zones (fig. 4). There, we report the location of single rock types within the aforementioned fault zone architecture. Since the samples were collected from a number of fault zones characterized by dissimilar attitude, kinematics, dimensional properties (ie., length, displacement, width of single structural domains, etc.), we do not consider the precise distance of single samples from main slip surfaces. As reported above in a reply to a comment provided by the reviewer 1 "This choice was meant to avoid any possible bias in the data interpretation provided by the intrinsic properties of single fault zones. For this reason, the 3D representation is scale-less, and does not include any reference related to the dimensional properties of single structural domains. However, as noted by the referees, we add information regarding the latitude/longitude of single samples in the appendix."

RC1: Also, the discussion section should be emphasized. The discussion section of the current manuscript has four short summaries, which seem to be very detailed, but may have some problems. First, some contents are said and left out, and the discussion is not deep enough, such as 6.1 Deformation mechanisms is a good direction, but the content is too simple. Secondly, the discussion is limited to the direct analysis or simple extension of the experimental results, and the overall discussion lacks the part of the actual impact, which mainly stays at the level of the results of this experiment, which is a little bit unfortunate. Consideration should be given to exploring the impact of the results on carbonate reservoirs, such as potential applications for geothermal energy development.

RE: The discussion direction indeed was oriented on the poro-permeability relationship and the different workflows adopted. The deformation mechanisms analysis is beyond our scope, but we can certainly integrate the discussion toward this direction as well as the impact for results on carbonate reservoirs and further applications.

RC1: Finally, there are some minor problems in the manuscript, such as problems with the citation format of the references. For example, line 100, MODIFIED FROM (Bourli et al., 2019). Please standardize the citation of references throughout the manuscript.

RE: We thank the reviewer for the comment and we have fixed the citation of the references.

---

## Author Comment (AC2)

RC2: The manuscript by Vinciguerra *et al.* comprises an experimental characterization of petrophysical properties, including density, porosity, Vp, Vs, electrical resistivity and microstructural analysis of carbonates and fault breccia in western Greece, with the aim to assess the porosity and permeability properties. The manuscript's combined approach is of potential interest to EGUsphere readers but several issues must be addressed before it can be accepted for publication. Below I list some general comments and suggestions that might strengthen the authors' interpretations of their results and several specific comments to be addressed when the authors revise the manuscript.

RE: We thank the reviewer for acknowledging the interest of our paper for EGUsphere and for providing valuable suggestions to improve our manuscript.

RC2: The main problem of the article is that the results and interpretations are presented quite segmented or isolated in a style of a working package results in a project and not in a synthesized overview, trying to link all the results of different analyses (analytical-experimental-modeling) and the uncertainties raised, in a complete interpretive model that would provide further recommendations for a proposed protocol that can be applied in similar case studies. In such an analysis, data don't always need to be fitted perfectly in a graph, and even when they don't, most important is to decipher and discuss the underlying major factors that cause weak or non-systematic relationships.

RE: As highlighted by the reviewer, by integrating three distinct protocols involving a variety of different analyses (analytical-experimental and modelling), we have provided a comprehensive and interpretative model to assess rock permeability and the porosity-permeability relationship. The validity and limitations of each specific approach has been thoroughly discussed, as well as the cause of weak or non-systematic relationship among the selected parameters. Moreover, the findings of each methodology adopted were compared and discussed in the dedicated sections, with the aim of providing general overviews and deciphering the physical properties of the tested materials.

RC2: Furthermore, they don't show the broader implication of their interesting approach with a comprehensive protocol for petrophysical carbonate analyses to adopt on similar case studies that can relate to potential applications on CO2 storage or geothermal energy as they state in their introduction.

RE: In our opinion, the results of our study provide a protocol of investigative tools that can be employed for case studies similar to those mentioned by the reviewer. Of course, the specific results are dataset dependent. However, the goal of this research is to assess a working protocol that can be used for the analyses of fractured and faulted carbonates, which are potential sites for CO2 storage, geothermal energy production, and many other applications, such as groundwater management, hydrocarbon recovery, pollutant transport in the subsurface. We will emphasize these aspects more clearly in the discussion section of the revised manuscript.

RC2: It is well documented that carbonate rocks are highly heterogeneous and have undergone complex sedimentation under various depositional environments, affected by post-depositional diagenetic processes and tectonism and for all these reasons they develop a variable and complex pore type network with complicated combination patterns (vugs, fractures etc.) which create intricate pore-throat structures. The complex pore-throat structure leads to the complexity of poro-perm relationship. Commonly, for such analysis as presented in this study, except for thin section DIA and UPV measurements, there should be additional analysis of SEM images, mercury intrusion or μ-CT scan images, because it would give a much better insight on the 3D pore structure than the 2D-image analysis they performed.

RE: One of the goals of our research was to distinguish between crack porosity (elongated, soft pores) and primary porosity (sub-rounded, stiff pores). To achieve this, a variety of methods were employed to carefully asses the pore type, geometry, dimension, and multiscale distribution within the selected samples. As acknowledged by the reviewer, imaging was used for this aim and integrated with the experimental analyses to provide further evidence, helping to better constrain the interpretation of the petrophysical data. A detailed analysis of the 3D pore structure was beyond the purpose of this study, which is why we did not carry out detailed microstructural analyses at the single-pore scale (cf. the SEM and μ-CT imaging analyses suggested by the reviewer).

RC2: Even reaching the end of the manuscript I am not quite convinced which are the main controlling factors of the weak poro-perm relationships. Are the different pore-throat structure types and their size that were not classified in detail? In this paper there is a lack of classification of the different pore-throat types. In order to have more reliable results for the poro-perm relationships there is a need to classify the different types of pore structure, their frequency distribution of pore- throat radius, (which can be multimodal, bimodal, centralized unimodal or asymmetric bimodal) and their connectivity. So effective porosity determination is crucial and is related to the determination of interparticle versus vuggy pores. Sometimes vuggy pores are evident, while other times are subjective and contain much uncertainty. In this study classification of vuggy pore space is not applied or described in more detail. Are there separate vugs interconnected only through the interparticle porosity or are touching vugs that form an interconnected pore system independent of the interparticle porosity? (see Lucia 1983). Thus pore segmentation is the first and also the key step for permeability estimation based on thin sections.

RE: The aim of the paper is not to classify the vuggy pore space structure, but rather to discriminate whether crack porosity or void space porosity are the more efficient carrier in terms of rock permeability. Given the presence of fault rocks and the high variety of carbonate lithologies and degrees of cementation, any classification would be lithology-dependent and, as such, would not contribute to the scope of the study. Other studies have already been carried out on carbonate fault rocks (Ferraro et al. 2018, 2019, 2020) regarding their vuggy pores structure associated with specific cementation types and measured petrophysical and ultrasonic properties. For this reason, we discuss our original data in light of the existing bibliography to better decipher the control exerted by primary (ie., stiff, sub-rounded pores) and secondary porosity (ie., soft, elongated pores) on the measured petrophysical properties of the studied carbonate rocks.

RC2: Another source of error may be the effective porosity determination which is related to pore size, since in images of thin sections obtained with optical microscopy, pores smaller than ~10 μm are difficult to resolve. This is relative of course to the size of the dominant pore system in the samples. For a sample with dominant pore system of 50–100 μm or larger, smaller than 10 μm pores are either intraparticle pores or small interparticle pores, both of which do not significantly contribute to permeability but for a sample with smaller dominating pores (<50 μm), the inability to resolve pores smaller than 10 μm that can be part of the dominant pore network will lead to permeability underestimation. Thus, the effective porosity is more relevant to permeability than the total porosity. In contrast, poorly-connected pores, mostly vuggy pores, may account for substantial porosity, but do not affect permeability significantly thus should not be counted in effective porosity and need be excluded for permeability calculation. Consequently, only effective pores, which are interpreted to be interparticle pores in carbonates, should be included in both 2D and 3D permeability simulation, as several studies suggest. Those issues and uncertainties should be discussed at least in their analysis.

RE: We agree with the reviewer that smaller connected porosity could potentially underestimate the permeability when using 2D image analyses. However, this should not significantly affect the total

permeability, as the connected macroporosity would primarily control the bulk permeability. This is why, in the present study, we used 3 different workflows: 2 of these workflows consider the 3D bulk properties assessed after lab measurements and modelling, while the third workflow deals with the properties deciphered after microstructural analysis of the study samples. The results from these 3 different workflows are fully discussed in the Discussion section and are all consistent with a microporosity underestimation effect, which can be considered negligible for the study hand specimens.

RC2: So, more information is needed about the pore-size distribution (PSD) and pore-throat types for each domain (Host rock-fractured host rock-breccia zone of fine breccia FZ) and which can be classified in different classes, as either cavity-fracture type, cavity type, pore-fracture type or simple pore type, and examined if each one type can be related to different poro-permeability relationship, instead of trying to fit all that in one best-fit line or curve.

RE: As discussed above, we believe that information on PSD, even if valuable, is not essential for the aim of our study. Once again, we highlight that the scope of this paper is to assess a working protocol that can be used for fractured and faulted carbonates exposed at the Earth's surface, and therefore affected by weathering processes.

RC2: Additionally, there is not much information about the various types of the diagenetic processes that these two main units (Senonian Limestones and Vigla limestones) have experienced, (i.e. early dolomitization or not, dissolution, cementation, etc.) which can have great influence in the pore-throat structure and consequently to the poro-perm relationship. Therefore, the authors should further examine and discuss the control of the various pore-throat types on the porosity- permeability relationship in order to establish a more reliable permeability calculation model.

RE: We are not dealing with the diagenesis of the units (for which we have provided the relevant references) or the nature of the pore-structure. In fact, previous works by Bourli and co-authors are cited in the reference list, and widely employed in the Discussion of our original data. As the reviewer correctly pointed out, the aforementioned approaches are certainly useful for fully assessing the diagenetic evolution of single carbonate units, but would be redundant given the existing bibliography and, therefore, are beyond the purpose of our study.

RC2: Sampling uncertainties- Since those cubic samples only cover a small proportion of the reservoir study interval, this should be defined in which interval these measurements refer to. There is no field image showing how the samples were taken, in which Senonian or Vigla unit interval, how the samples are distributed along the brecciated fault zones and the sample locations on map or table with coordinates. Field images are not included to provide the reader with an objective overview of where the samples were collected. In such kinds of studies, it is important to provide some images of the outcrops, or at least representative ones in figs and a complete photo list of samples in a supplementary material file.

RE: The reviewer raises a common issue in experimental studies: the representativeness of the study samples. This is crucial to consider. For this reason, in fig. 4, we provide a synoptical 3D representation of the main structural domains assessed for the study fault zones, and the location of single rock types within the aforementioned fault zone architecture. Since the samples were collected from a range of fault zones characterized by dissimilar attitude, kinematics and dimensional properties (ie., length, displacement, width of single structural domains, etc.), we did not consider the precise distance of each sample from the main slip surfaces. This choice was made to avoid any possible bias in the data interpretation provided by the intrinsic properties of single fault zones. Consequently, the 3D representation is scale-less, and does not include any reference to the

dimensional properties of the individual structural domains. However, as suggested, we will add information regarding the latitude/longitude of each sample in the appendix.

RC2: The interpretation of data in parts is not well presented or explained, and there is no correlation or at least discussion of their results with prior studies on poro-perm relationships in carbonates (e.g from Middle East or China etc), which could enhance the broader interest of the scientific community. As it stands now, it seems more like a site specific project outcome, which unlikely can give an insight or workflow protocol for a future study in other carbonate units.

RE: Any specific outcome from such an integrated approach is inevitably site-specific. However, we have compared and contrasted our findings with the existing dataset for carbonate fault rocks exposed along peninsular Italy (cf. Ferraro et al., 2018, 2019, 2020) and proposed a valid protocol for the petrophysical analysis of fault zones in carbonates. Again, we emphasize that the scope of our work is NOT the characterization of single sedimentary units (as said, Bourli and co-authors fully documented these units) but the detailed analysis of the fractured and faulted carbonate samples collected form the single structural domains (cf. the previous reply).

RC2: Some parts of the analyses are not quite clear for what purpose they were conducted and what was the benefit for their analysis: for example the fractal dimension measurement and how significantly impacts the goals of this study should be mentioned. Except the quantitative aspect of it, what information can further give? In my point of view, fractal dimension can provide an indication of pore-surface complexity and scaling behavior of the object but this could be more reliable if it involves image analysis of SEM-BSE or results from m-CT 3D reconstruction of dense slices of images. In the same sense, the resistivity results were very surficially integrated to the rest of the results.

RE: Fractal dimension was used to assess the distribution of selected objects (clasts and survivor grains for rock textural analysis, pores for 2D void space analysis) within the selected images. We outline how the precise 3D characterization of the primary and secondary pores was not the scope of this work, which on the contrary focused on the validation of a working protocol to be used for the petrophysical analyses of fractured and faulted carbonate rocks.

RC2: Further minor comments on text are listed below:

Line 21- textural anisotropy? What do authors mean by that? They assume that in different orientations the texture is different or they mean heterogeneity?

RE: We assume that orientation of the texture (i.e. rock matrix plus voids space) is different, as evidenced from the physical properties.

RC2: Line 22- imply that these selected samples are from different structural depths?

RE: No, we imply that samples properties evolve at depth, this is why we present also measurements at increasing pressure.

RC2: Often there is a wide variety of aperture width along the microfractures. From the studied thin sections is there any estimation of the % of the completely healed, partially filled or open microfractures?

RE: No, we actually did not perform such a diagenetic analysis. However, we note that the mineralization documented along coated slickensides and veins was analyzed in detail by mean of

integrated geological, structural, mineralogical and geochemical analyses, and published in the Tectonics journal (Smeraglia et al., 2023) as reported in the Reference list.

RC2: Line 54- non-cohesive

RE: Ok, the proposed modification is accepted. We thank the reviewer for the comment.

RC2: Line 82- more relevant references should be referenced for the thrusting tectonics in FTB of Hellenides than the introductory article of Robertson and Dixon 1984, which deals mostly with ophiolite displacement and microplate tectonics in the eastern Mediterranean. In general, the Geological setting section needs an update and repolishing in references. There is nothing for the subsurface evaporitic diapirism that seems to play a crucial role for the deformation in western Hellenides FTB. Also only short reference is given for the units of Senonian and Vigla formations and their internal deformation, which are the main sampling units in this study.

RE: We thank the reviewer for this comment. We acknowledge the complexity of the area and have cited the most significant publications that address the structural setting and tectonic evolution of the study area. However, we believe that all the most relevant publications have been included in the Reference list. The effects related to diapirism, which certainly contributed to the large-scale deformation style of the western Hellenides fold-and-thrust belt, are therefore not included in the manuscript. As mentioned previously, since this work DOES NOT assess the petrophysical variations within the single sedimentary units but FOCUSES only on the single structural domains, we believe that the provided references offer a sufficient framework to discuss our original data.

RC2: Line 111- samples in proximity refer to fault damage zone (internal or external) or not?

RE: No, they refer to host rock samples (cf. the HR acronym) collected in the vicinity (proximity) of the fault damages and hence OUTSIDE of the fault zones. These samples, after detailed field and microstructural observations, do not include any fault-related fracture and were considered in order to compare and contrast their petrophysical properties with respect to those assessed for similar samples (same sedimentary facies) collected farther away from the same fault zones. Results supported our preliminary interpretation, and hence confirmed that the samples in proximity to faults were not actually pertaining to the single fault damage zones.

RC2: Line 115- a cubic block taken normal to bedding comprises sections that are parallel to bedding strike-perpendicular to strike and parallel to bedding (x,y,z reference axes). Which sections were used for the 2D image analysis isn't very clear for the reader. So, more details should be given on how the samples were chosen and which orientations were studied.

RE: We thank the reviewer for this comment. The orientation of the single thin sections is mentioned in line 125. However, to provide the required information more clearly, we have added the following text "unless carefully reported, the samples collected from the fault zone were sectioned slip-parallel, while those taken away from the fault zones parallel to the strike of single beds"

RC2: Additionally, at least a stereonet should be added in the main map, showing the regional bedding orientations, the fault zones that are discussed in the text and the main fracture set orientations for the Senonian and Vigla units.

RE: Due to the great amount of information provided in the manuscript (18 different figures!), we refer to a previous work we carried out in the study area (Smeraglia et al., 2023) when reporting the

structural setting of the Araxos Promontory, and the fault-related fracture sets documented within single fault zones.

RC2: Comments in sampling method.

Why were macropores avoided? And from what size and above? They don't contribute to the total porosity? (see also previous comments on porosity above). Since there are no graphs for pore size distribution for the samples, which is the dominant pore-size range for each sample? That is an important issue when you exclude specific pore size ranges.

RE: The dimensions of pores assessed through 2D image analysis are reported in table 3, where all the required information for each sample is provided.

RC2: Saturation of samples were not performed with vacuum? (less than 800 Pa for an hour). For grain mass Ms samples shouldn't be dried? That I think refers to methodology of ISRM 1977, as far as I am aware.

RE: No, we saturated the samples at room pressure by leaving them immersed in water for one week. Afterward, we used a moistened cloth to dry only the surface of the samples, as recommended by ISRM 1977. For the determination of Ms, we chose to avoid drying the samples in an oven, as temperature up to 105 °C could induce surface cracking, as demonstrated by Vagnon et al. 2019). Therefore, we determined Ms before saturating the rock specimens.

Vagnon F. et al. 2019. Effects of thermal treatment on physical and mechanical properties of Valdieri Marble - NW Italy. International Journal of Rock Mechanics and Mining Sciences, 2019, 116, pp. 75–86.

RC2: Might be some issues with image analysis. See Fig. 3 bottom image pair where blue-epoxy pore space of two areas in bottom right are disconnected but in the bitmap image shown as well connected pores. Porosity measurements from images have a threshold limit, represented by the minimum detectable pore dimension. What is that limit in image analysis? What is the pixel size of thin-section images?

RE: Unfortunately, we disagree with the reviewer. The single thin sections were carefully analyzed under the optical microscope before selecting the areas for subsequent digital image analysis. Regarding the images mentioned by the reviewer (Fig. 3 bottom image), the two areas in the bottom right are actually connected, as the black spot in between refers to a drop of glue on the thin section. For this reason, the two area were merged in the binary image on the right. Finally, all thresholds and pixel sizes are reported in the text.

RC2: In 3.3.2. UPV section, what are the relative errors of the velocity measurements? Commonly a PCA analysis is done of density data and UPV values reordered along x,y,z direction to see if there are representative results for each block.

RE: The error bars in UPV measurements account for errors related to sample dimensions and the peaking of the first arrival time of the ultra-sonic wave generated along the single directions. We also perform error propagation to determine whether any observed heterogeneity is a genuine variation or falls within the natural variability of the material.

RC2: Line 268- something is missing from the sentence, please check.

RE: We have shortened and rephrased the sentence as: Since the time decay α is function of i) permeability, ii) viscosity of the pore fluid; iii) the geometry of the sample; iv) the storage of the upstream (Cu) and downstream (Cd) reservoirs, permeability values can be calculated via α for each step of increasing pressure.

RC2: Figure 4- samples should be rearranged in the figure either from FZ to the host rock or opposite and not mixed up (figs 4a-f). A scale or distance should be shown in the block model of the sampling area. Field image views of the outcrops that samples were collected from the fault zone and crush breccia zone would be valuable to add (either together with Fig.4 or in a Supplementary material section (together with sample coordinates)

RE: We thank the reviewer for the comment. Regarding the arrangement of figure 4, we change the figure accordingly. Regarding the distances of individual fault rock samples from the main slip surfaces, we did not report them for the reasons explained above.

RC2: Table 1 needs a legend explaining what is bis, pento a, pento b etc. AR 43 having identical % clast and % of matrix in both orientations seems a bit peculiar.

RE: The suffixes bis, penta a, b, ecc. refer to thin sections obtained from single hand specimens. Regarding the results obtained for the sample AR 43 are actually correct. We double checked the output of our digital image analysis, and confirm the aforementioned values.

RC2: Line 307- is 2D optical porosity

RE: Ok

RC2: Lines 308-329- all these data should be easier visualized if added in table 1 , showing which samples are from Senonian or Vigla, if they include stylolites, veins or fractures, porosity values and Do(pore) values.

RE: These data cannot be added to the table 1 because refer to properties assessed after thin section observations and digital image analyses (please cf. reference to individual figures reported in the text).

RC2: Line 319- which sample is the fractured packstone?

Matrix and cement are considered the same here but are different in terms of the diagenetic history. What part (%) consists of the fine grained material and what part (%) is the binding cement material?

RE: We thank the reviewer for this comment. In order to fulfill the request, we add all the information obtained after digital image analysis of the pore space in the appendix.

RC2: Authors mention pores aligned to veins? Or do they mean microfractures connecting moldic pores?

RE: Line 330: "The samples include numerous microfractures and veins. Pores are mainly aligned along the former structural elements." By reporting these sentences we mean that the pores are aligned along the microfractures.

RC2: Line 354- normalization was performed with the formula xnormalized= (x-xmin)/range of x and then get the average?

RE: No, in order to make a direct comparison, each value was normalized by dividing the considered parameter by the maximum value of that parameter, and then averaging the results.

RC2: Fig.6 I cannot see so clearly that trend which authors describe here. Most of the samples in FHR-FZ show higher density than HR-PFZ or HR-AFZ. This should be discussed further. Only CFB-FZ show significant decrease in density

RE: There is a statistical trend, where CFB-FZ clearly plays a major role. We can remove the trends if the reviewer believes it would be better.

RC2: Figs 6-7. A relative or average distance between main fault slip zone and rest of the HR domains should be shown. It comprises 10's of meters or less? Are there any important shear zones in the FHR , CFB domains?

RE: As already stated above, the true dimensional properties and distances of single structural domains from the main slip surfaces are not provided, as these values vary significantly depending on the attitude, kinematics, length and amount of displacement of each fault zone.

RC2: What are the differences of UPV measurements along the 3-orthogonal directions and how we can detect anisotropy along a specific orientation if all measurements are averaged for each sample? You discuss that in lines 394-395 but since you averaged all 3 directions this can't be validated.

RE: We did not include the figure for Vp (or Vs) in order to reduce the number of figures, and because this information can be retrieved (even partially, we agree) from Figure 10. If the reviewer believes it would be better to provide this figure as well, we can include it as supplementary material.

RC2: There is no information how the microfracture orientations are aligned with any of the 3-axes. Later in text, using equation 16 to calculate crack density requires cracks randomly oriented and distributed. I am not quite sure if your samples meet the conditions for perfect isotropic configuration.

RE: No, we cannot assess the microfracture orientation, but we can evaluate the relative anisotropy between the different directions. In many samples, the anisotropy is not strongly marked, and consequently, the assumption of isotropic behavior—or, more accurately, randomly oriented fractures—is not entirely incorrect. Moreover, even in samples with more pronounced anisotropy, there is no clear orientation of the cracks, which justifies the use of Equation 16.

RC2: Line 435- Cementation factor plays a more important role in the UPV and permeability relationship?

RE: It does, as discussed.

RC2: Line 491- $\zeta$ can't be aspect ratio and crack density also, something is labeled wrongly here. (or one is in italics and causes confusion?)

RE: We apologize for the mistake. $\zeta$ refers to the aspect ratio, and $\rho$ refers to the crack density.

RC2: In Discussion, authors mention '' the main mechanisms of samples experienced during deformation phases''. Which are these deformation phases they refer to?

RE: The sentence refers to the general deformation that the samples have undergone throughout their geological history. We were referring to the various deformation phases they experienced over time, which include, but are not limited to, compression, shear, and extension processes.